# Nested circuits mediate the decision to vocalize

**Shuyun Xiao[†‡], Valerie Michael[†], Richard Mooney\***

Department of Neurobiology, Duke University, Durham, United States

**Abstract** Vocalizations facilitate mating and social affiliation but may also inadvertently alert predators and rivals. Consequently, the decision to vocalize depends on brain circuits that can weigh and compare these potential benefits and risks. Male mice produce ultrasonic vocalizations (USVs) during courtship to facilitate mating, and previously isolated female mice produce USVs during social encounters with novel females. Earlier we showed that a specialized set of neurons in the midbrain periaqueductal gray (PAG-USV neurons) are an obligatory gate for USV production in both male and female mice, and that both PAG-USV neurons and USVs can be switched on by their inputs from the preoptic area (POA) of the hypothalamus and switched off by their inputs from neurons on the border between the central and medial amygdala ($Amg_{C/M-PAG}$ neurons) (Michael et al., 2020). Here, we show that the USV-suppressing $Amg_{C/M-PAG}$ neurons are strongly activated by predator cues or during social contexts that suppress USV production in male and female mice. Further, we explored how vocal promoting and vocal suppressing drives are weighed in the brain to influence vocal production in male mice, where the drive and courtship function for USVs are better understood. We found that $Amg_{C/M-PAG}$ neurons receive monosynaptic inhibitory input from POA neurons that also project to the PAG, that these inhibitory inputs are active in USV-promoting social contexts, and that optogenetic activation of POA cell bodies that make divergent axonal projections to the amygdala and PAG is sufficient to elicit USV production in socially isolated male mice. Accordingly, $Amg_{C/M-PAG}$ neurons, along with $POA_{PAG}$ and PAG-USV neurons, form a nested hierarchical circuit in which environmental and social information converges to influence the decision to vocalize.

**\*For correspondence:**
mooney@neuro.duke.edu

[†]These authors contributed equally to this work

**Present address:** [‡]Department of Biology, Stanford University, Stanford, United States

**Competing interest:** The authors declare that no competing interests exist.

## Editor's evaluation

Vocalizations are controlled by neural circuits connecting the amygdala and periaqueductal gray. This study presents valuable measures of the neurons that suppress vocalization in appropriate contexts using a rich variety of behavioral, imaging, optogenetic, and tracing methodologies. The evidence supporting the claims of the authors is solid, and their results provide synaptic mechanisms by which vocal-promoting and vocal-suppressing signals could interact in the mouse's brain to underlie the hierarchical control of vocalization. The work will be of interest to neurobiologists working on motor control and vocalization.

## Introduction

Vocalizations help foster mating and social bonding in most mammals (*Seyfarth and Cheney, 2010*). However, vocalizing in the wrong context can increase the risk of attracting the attention of eavesdropping predators or social rivals (*Akre et al., 2011*; *Roberts et al., 2007*). Therefore, to make an adaptive vocal decision, the neural circuits that regulate vocalization must respond not only to affiliative social and sexual cues generated by conspecifics but also to threats and aversive social stimuli, weighing them against the potential benefits of mating and social interactions. How these various and often opposing types of sexual, social, and environmental factors are integrated in the brain to influence the decision to vocalize remains unclear. Here, we explored the idea that vocal-suppressing

neurons in the amygdala encode information about predators and threatening rivals, and that they influence the decision to vocalize by weighing this information against that about potential social partners and mates.

Mice produce ultrasonic vocalizations (USVs) to communicate with conspecifics (*Chabout et al., 2015*; *Hammerschmidt et al., 2012*; *Maggio and Whitney, 1985*; *Neunuebel et al., 2015*; *Portfors, 2007*; *Tschida et al., 2019*; *Warren et al., 2018*; *Warren et al., 2020*; *Zhao et al., 2021*). As with the production of affiliative vocalizations in other mammals, USV production in mice is enhanced by certain social factors, including the presence of suitable sexual partners and previous experience with social isolation (*Chabout et al., 2015*; *Zhao et al., 2021*). For instance, it is uncommon for either male or female mice to vocalize when they are alone in the absence of any social stimuli (*Tschida et al., 2019*; *Michael et al., 2020*). When resident adult males are presented with male intruders in their home cage, they rarely vocalize and often initiate attack (*Lee et al., 2014*). Yet, when presented with females, male mice produce USVs robustly to females to facilitate mating (*Chabout et al., 2015*; *Neunuebel et al., 2015*; *Portfors, 2007*). In female mice, the context and function of robust vocalizations are less elucidated. Nevertheless, a recent study by *Zhao et al., 2021* showed that after being subjected to social isolation, resident female mice start to produce significantly more USVs in response to a novel female, although the cues that drive these female-female vocalizations and their social function are not well understood (*Hammerschmidt et al., 2012*; *Zhao et al., 2021*).

The recent identification of neurons in the midbrain periaqueductal gray (PAG) that serve as an obligatory gate for USV production (PAG-USV neurons) provides the potential for understanding the circuit logic underlying a mouse's decision to vocalize (*Tschida et al., 2019*). In fact, a GABAergic projection from the hypothalamic preoptic area (POA) to the PAG drives USV production in male and female mice in part through disynaptic disinhibition of PAG-USV neurons (*Chen et al., 2021*; *Michael et al., 2020*). Conversely, a subpopulation of GABAergic neurons on the border of the central and medial amygdala (Amg$_{C/M-PAG}$ neurons) directly inhibits PAG-USV neurons, and optogenetically activating these neurons in male mice suppresses their courtship USVs (*Michael et al., 2020*). Such convergent and opponent circuit architecture points to the PAG as one site where vocal-promoting signals could be weighed against vocal-suppressing signals to influence the decision to vocalize. However, while POA neurons that express Estrogen receptor alpha (Esr1), which is a molecular marker of POA neurons that innervate the PAG-USV region (*Chen et al., 2021*; *Michael et al., 2020*), are strongly activated in male mice during female courtship (*Chen et al., 2021*), the extent to which Amg$_{C/M-PAG}$ neurons are excited by predators or social rivals remains unknown. Thus, one question we sought to answer is whether Amg$_{C/M-PAG}$ neurons are activated by predator cues or during social contexts that typically suppress USV production in male and female mice.

Moreover, carefully weighing the potential costs and benefits of vocalization is likely to involve integration at other sites beyond the PAG. Indeed, some POA neurons that project to the PAG-USV region also project to the region of the amygdala that contains Amg$_{C/M-PAG}$ neurons (*Michael et al., 2020*), raising the possibility that Amg$_{C/M-PAG}$ neurons integrate USV-promoting as well as USV-suppressing information. However, whether POA neurons that provide input to the Amg$_{C/M}$ are active during USV-promoting social encounters and whether activity in this subset of POA neurons is sufficient to drive USV production in the absence of social cues remain untested. Lastly, the physiological properties of synaptic inputs from the POA onto Amg$_{C/M-PAG}$ neurons are unknown, but these properties are important to understand whether the Amg$_{C/M}$ balances competing types of information to determine whether a mouse will vocalize.

To this end, here we investigated the role of Amg$_{C/M-PAG}$ neurons and their inputs from the POA in regulating USV production using in vivo fiber photometry of identified cell groups, anatomical and functional circuit mapping, and functional manipulations of neuronal activity in socially interacting and isolated mice. These experiments reveal that Amg$_{C/M-PAG}$ neurons in both male and female mice are strongly activated by cues or contexts that suppress USV production, and that optogenetically activating Amg$_{C/M-PAG}$ neurons selectively suppresses USV production in both male and female mice across a variety of social contexts in which they typically vocalize. To further explore how vocal promoting and vocal suppressing drives are weighed in the brain to influence vocal production, we conducted an additional series of experiments in male mice, where the drive and courtship function for USVs are better understood. We found that Amg$_{C/M-PAG}$ neurons receive monosynaptic inhibitory input from POA neurons that also project to the PAG, that these inhibitory inputs are active in USV-promoting social

contexts, and that optogenetic activation of POA cell bodies that make divergent axonal projections to the amygdala and PAG is sufficient to elicit USV production in socially isolated male mice. Together these experiments reveal a nested hierarchical circuit in which environmental and social information is integrated and weighed at multiple processing steps to influence the decision to vocalize.

## Results

### Amg$_{C/M-PAG}$ neurons are active in response to threatening stimuli

Given that optogenetically activating Amg$_{C/M-PAG}$ neurons suppresses USV production in male mice during courtship encounters with females, we reasoned that Amg$_{C/M-PAG}$ neurons would be highly active during threatening contexts in which vocalization is potentially risky for the sender and thus likely to be suppressed. First, we tested how USV production during social encounters was affected by the synthetic fox urine odor 2-methyl-2-thiazoline (2MT), a derivate of 2,5-dihydro-2,4,5-trimethylt hiazoline (TMT) (*Day et al., 2004*; *Lin et al., 2006*; *Root et al., 2014*), a component of fox urine that is highly aversive to mice. In comparison to TMT, 2MT has been reported to induce freezing behavior more reliably (*Bruzsik et al., 2021*; *Isosaka et al., 2015*). We measured the number of USVs produced by male or previously isolated female (see following section) mice during a 5-min encounter with a novel female conspecific in the presence or absence of 2MT. We found that both male and female mice produced significantly fewer USVs when encounters with a novel female occurred in the presence of 2MT compared to controls (*Figure 1A*; *Figure 1—video 1*; *Figure 1—video 2*).

Next, we used an intersectional viral strategy to express the calcium indicator GCaMP8s in Amg$_{C/M-PAG}$ neurons (*Figure 1B*) and used fiber photometry to measure calcium signals in these neurons in male and female mice as they investigated a dish containing either 2MT, a control odor (ethyl tiglate), a novel plastic 'toy', or the soiled bedding of male or female conspecifics (*Figure 1C*). The rationale for conducting fiber photometry experiments with only social partners, *or* novel objects, *or* 2MT (*Figure 1C*) as opposed to repeating the behavioral paradigm used in the vocalization behavioral test described above (*Figure 1A*) is that we sought to isolate Amg$_{C/M-PAG}$ activity patterns that were evoked by one stimulus type or another, rather than a combinatorial signal that would be difficult to interpret in the absence of these single condition data. In both sexes, calcium signals in Amg$_{C/M-PAG}$ neurons were significantly elevated above baseline when the animal approached and investigated dishes containing any of these items (*Figure 1—figure supplement 2*), but they were the highest when investigating the dish containing 2MT (*Figure 1D–F*). The overall pattern of responsiveness to these various stimuli was similar between male and female mice (*Figure 1F*; *Figure 1—figure supplement 1A*). Furthermore, the calcium increase evoked by 2MT was not linked to subsequent bouts of freezing, suggesting that activity in these neurons was driven by detection of the odorant rather than the subsequent behavioral response (*Figure 1—figure supplement 1B*). Therefore, an innately aversive predator odorant suppresses USV production and strongly activates USV-suppressing Amg$_{C/M-PAG}$ neurons in both male and female mice.

We then tested how Amg$_{C/M-PAG}$ neuron activity was affected during social interactions with other conspecifics that can either promote or suppress USV production. We used fiber photometry to measure calcium signals of Amg$_{C/M-PAG}$ neurons in freely behaving male and female mice during 5-min encounters with novel male and female conspecifics. These encounters resulted in non-aggressive interactions when USV production was more likely to occur or aggressive interactions (i.e. fights between two males) when USV production is absent (we confirmed that males did not produce USVs during fights). In both sexes, we observed similar small increases in Amg$_{C/M-PAG}$ neuronal activity when calcium signals were aligned to the onsets of non-aggressive social encounters with male or female conspecifics, or with a mouse's investigation of a novel object (*Figure 1G*; *Figure 1—figure supplement 1A*; *Figure 1—figure supplement 2A*). However, calcium signals of Amg$_{C/M-PAG}$ neurons did not change significantly at the onset or offset of USV bouts, suggesting that Amg$_{C/M-PAG}$ neurons are not simply turning off USV bouts during non-aggressive social encounters (*Figure 1H*). In contrast, Amg$_{C/M-PAG}$ neuronal calcium signals increased markedly during aggressive interactions between two males that included attacks (*Figure 1I–J*). Although many of these epochs included aggressive behavior from both males, Amg$_{C/M-PAG}$ neuronal calcium signals were larger in the male being attacked (*Figure 1—figure supplement 1C*). In summary, Amg$_{C/M-PAG}$ neurons exhibit large increases in activity during aggressive social encounters when USVs are not typically produced and much smaller increases in

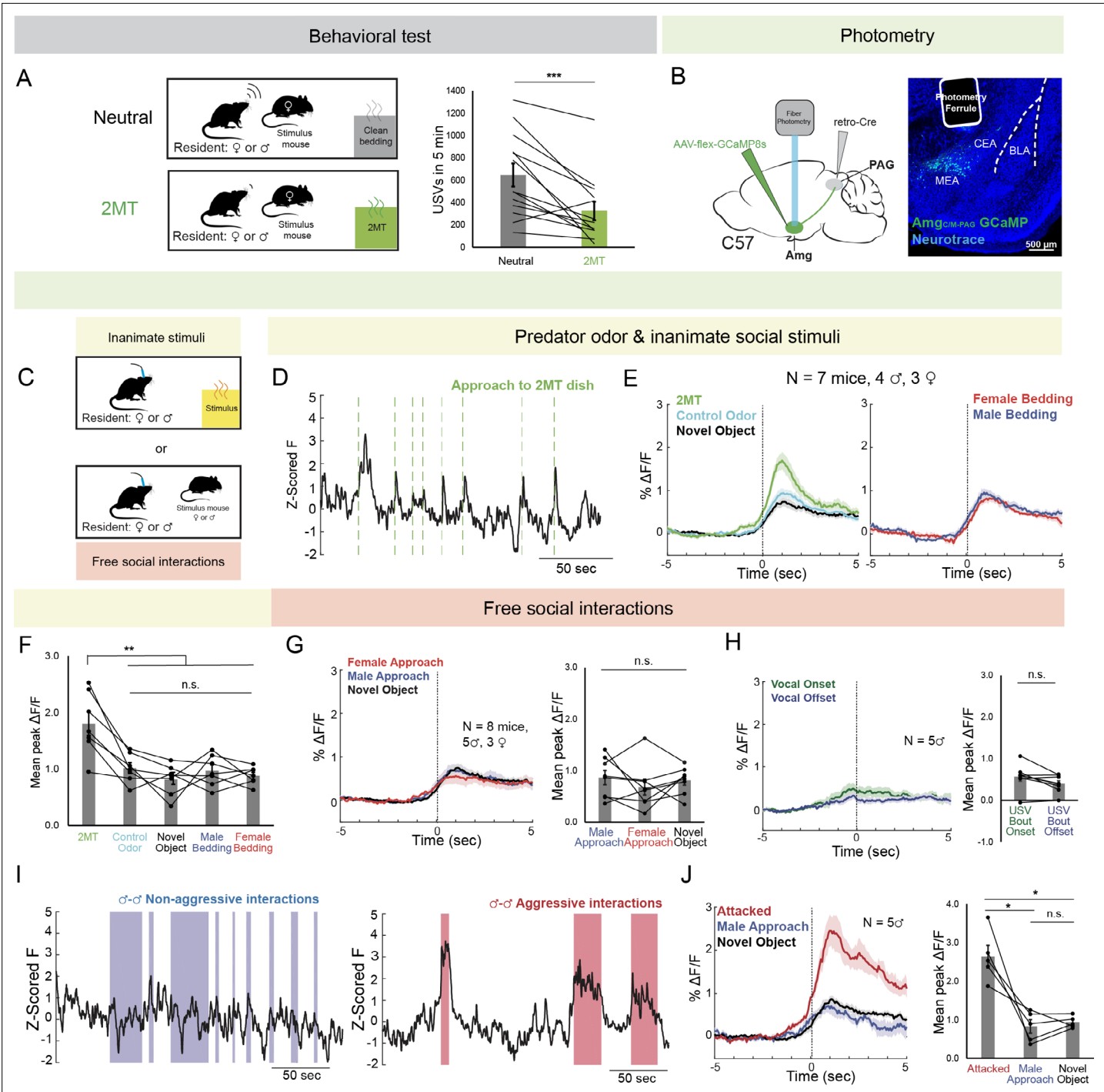

**Figure 1.** Amg_{C/M-PAG} neurons are active in response to threatening stimuli. (**A**) (Left) Schematic shows the behavioral paradigm: in the neutral condition, a stimulus female mouse along with a dish containing clean bedding was placed into the home cage of either a male or a female resident; in the 2MT condition, a stimulus female mouse along with a dish containing 2MT was placed into the home cage of either a male or a female resident. (Right) The number of USVs produced by male or previously isolated female mice during a 5-min encounter with a novel female conspecific in the presence or absence of 2MT. (N=13, 11 males, 2 females; ***p<0.001; paired *t* test). (**B**) (Left) Viral strategy for photometry recording of Amg_{C/M-PAG} neurons. (Right) Confocal image showing photometry ferrule tract and GCaMP8s expression in Amg_{C/M-PAG} neurons. (**C**) Schematic showing behavioral paradigm during photometry recordings: unlike the behavioral test shown in (**A**), here only one stimulus was presented at a time. (**D**) Calcium signal in Amg_{C/M-PAG} neurons as a male mouse approached and investigated a dish containing 2MT. Dashed green lines indicate approach onset. (**E**) Average Amg_{C/M-PAG} activity in males (N=4) and females (N=3) as the animal approached and investigated a dish containing either 2MT, a control odor (ethyl tiglate), or a novel plastic 'toy'. (**F**) Mean peak Amg_{C/M-PAG} activity when approaching a dish containing either 2MT, a control odor (ethyl tiglate) or a novel plastic 'toy' or the soiled

*Figure 1 continued on next page*

*Figure 1 continued*

bedding of male or female conspecifics (N=7, 4 males, 3 females; **p<0.01; n.s. p>0.05; one-way ANOVA followed by post-hoc pairwise Tukey's HSD tests). Error bars represent S.E.M. (**G**) Average (left) and mean peak (right) Amg$_{C/M-PAG}$ activity in males (N=5) and females (N=3) during non-aggressive social encounters with male or female conspecifics, or with a mouse's investigation of a novel object (n.s. p>0.05; one-way ANOVA followed by post-hoc pairwise Tukey's HSD tests). Error bars represent S.E.M. (**H**) Average (left) and mean peak (right) Amg$_{C/M-PAG}$ activity aligned to vocal onset and offset (N=5 males; n.s. p>0.05; paired *t* test). (**I**) Calcium signal in Amg$_{C/M-PAG}$ neurons during non-aggressive (left) and aggressive (right) male-male interactions. (**J**) Average (left) and mean peak (right) Amg$_{C/M-PAG}$ activity in males (N=5) during attack, non-aggressive social encounters with male conspecifics, or with a mouse's investigation of a novel object (*p<0.05; n.s. p>0.05; one-way ANOVA followed by post-hoc pairwise Tukey's HSD tests). Error bars represent S.E.M.

The online version of this article includes the following video, source data, and figure supplement(s) for figure 1:

**Source data 1.** Source data for *Figure 1*, *Figure 1—figure supplements 1 and 2*.

**Figure supplement 1.** Amg$_{C/M-PAG}$ neurons are the most strongly activated by threatening stimuli.

**Figure supplement 2.** Amg$_{C/M-PAG}$ neurons respond to various stimuli.

**Figure 1—video 1.** Robust social vocalizations in the presence of neutral odor.

https://elifesciences.org/articles/85547/figures#fig1video1

**Figure 1—video 2.** Predator odor (2MT) suppresses social vocalizations.

https://elifesciences.org/articles/85547/figures#fig1video2

activity during affiliative social encounters in which USVs frequently occur. These observations support the idea that Amg$_{C/M-PAG}$ neurons actively suppress USV production in the presence of threatening stimuli.

## Activating PAG-projecting Amg$_{C/M}$ neurons suppresses female USV production

While we previously showed that optogenetic activation of Amg$_{C/M-PAG}$ neurons transiently suppressed courtship USVs in male mice without disrupting non-vocal social behavior (*Michael et al., 2020*), it remained an open question as to whether these neurons also suppress USV production in female mice. To resolve this issue, we adopted a recently developed behavioral protocol that can reliably elicit robust USV production from female mice (*Zhao et al., 2021*). To elicit high numbers of USVs, female mice were socially isolated for at least three weeks and then, on the day of testing, a novel female conspecific was introduced to the socially isolated experimental animal's home cage (*Figure 2A*). We observed that while group-housed female mice produce few USVs in response to a novel female conspecific, previously isolated female mice produced significantly more USVs per minute, more USV bouts per minute, and longer USV bouts, and in fact vocalized at rates comparable to isolated male mice during female presentation (*Figure 2A*). This finding confirms the recent study by *Zhao et al., 2021*, and provided us with a useful entry point for addressing whether Amg$_{C/M-PAG}$ neurons suppress USV production in female mice.

To this end, we used an intersectional viral strategy to selectively express channelrhodopsin (ChR2) in Amg$_{C/M-PAG}$ neurons of female mice (*Figure 2B*). We found that this strategy labels neurons in the Amg$_{C/M}$ but not the CeA of female mice, similar to the pattern previously observed with this strategy in male mice (*Figure 2B*; see also Figure 4A in *Michael et al., 2020*). Using the social isolation protocol, we were able to elicit high numbers of USVs routinely in 5 of 7 Amg$_{C/M-PAG}$-ChR2 female mice as they investigated a novel female partner. Optogenetic stimulation of Amg$_{C/M-PAG}$ neurons in freely vocalizing female mice immediately suppressed USV production, and this suppression persisted throughout the duration of the laser pulse (*Figure 2C–D*, N=5 of 5 female mice). The optogenetically evoked decrease in USV rate was significant when compared to a light-only control condition in which the same mice were connected to a dummy ferrule that shone blue light above their heads, a control condition in which the same mice did not receive laser stimulation and USV rates decreased naturally, and to a separate group of female mice in which GFP was expressed in Amg$_{C/M-PAG}$ neurons and that were subjected to optical fiber illumination in the Amg (*Figure 2D*; *Figure 2—video 1*; N=5 female mice in the optogenetic condition, light control condition, and natural bout decay condition and N=5 female mice in the GFP control condition, p<0.01 for differences between ChR2 group vs. control groups during laser time; two-way ANOVA with repeated measures on one factor, p<0.01 for interaction between group and time, followed by post-hoc pairwise Tukey's HSD tests). After laser

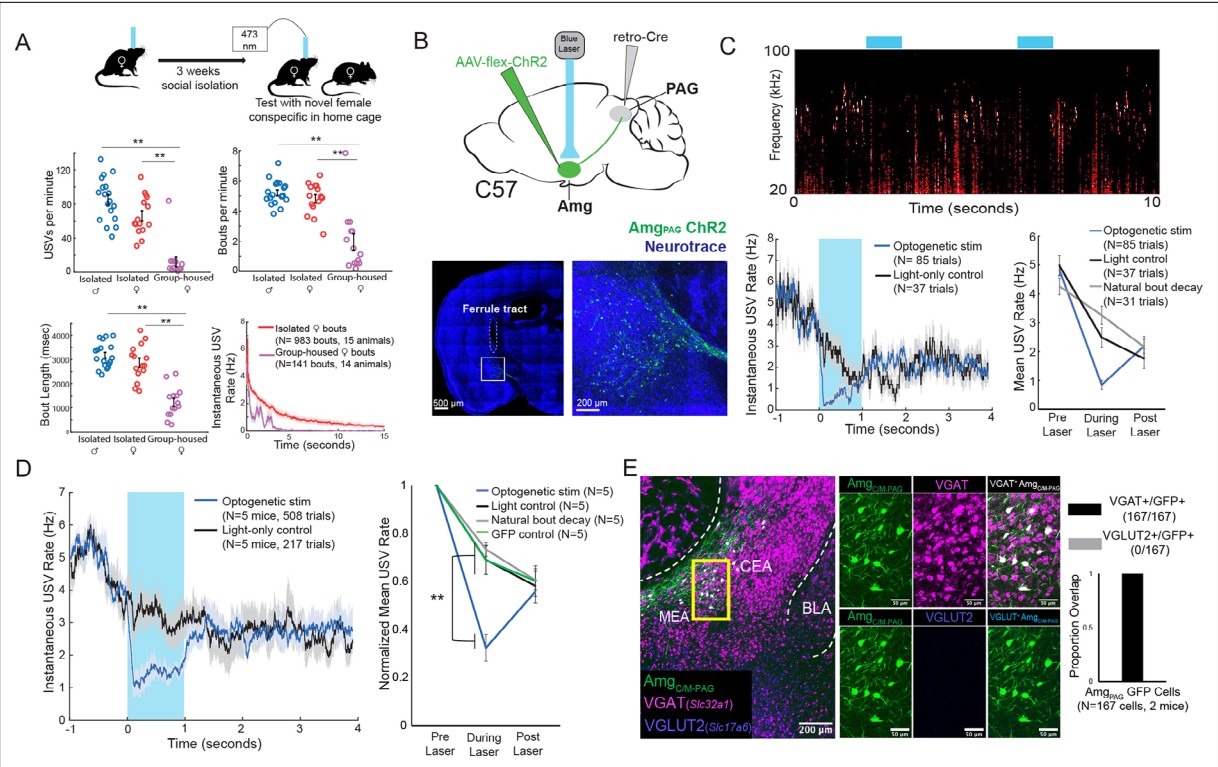

**Figure 2.** Activation of Amg$_{C/M-PAG}$ neurons suppresses vocalizations in female mice. (**A**) (Top) Behavioral paradigm for inducing USV production from female mice. (Bottom) Isolated female mice had similar numbers of USVs produced per minute, number of USV bouts produced per minute, and bout lengths when compared to isolated male mice, while group-housed females produced significantly fewer USVs per minute and significantly fewer and shorter bouts (N=18 isolated males, 15 isolated females, and 14 group-housed females, one-way independent sample ANOVAs found a significant effect of condition on USVs/min, USV bouts/min, and USVs/bout, post hoc pairwise Tukey's HSD tests found that for all three measures **p<0.01 for isolated males versus group housed females, **p<0.01 for isolated females versus group housed males, and no statistically significant difference between isolated males and isolated females). Error bars represent S.E.M. (**B**) (Top) Viral strategy for optogenetic activation of Amg$_{C/M-PAG}$ neurons in female mice (performed in N=5 females). (Bottom) Confocal image of representative Amg$_{C/M-PAG}$ cell body labeling and ferrule placement achieved with this strategy. (**C**) (Top) Example spectrogram showing a representative set of trials in which activation of Amg$_{C/M-PAG}$ neurons suppresses ongoing female USV production. (Bottom Left) Example traces of the USV rate during optogenetic stimulation vs a light-control condition for all trials (N=85 optogenetic trials and 37 light-only control trials) for a single representative animal. Error shading above and below the mean represents S.E.M. (Bottom Right) Quantification of the mean USV rate 1 s before laser stimulation, during a 1 s laser stimulation, and 1 s after laser stimulation (blue) compared to light-only control trials (black) and to the natural decay of vocal bouts with no stimulation (gray) for one representative animal. Error bars represent S.E.M. (**D**) (Left) Average USV rate traces during optogenetic stimulation and a light-only control for all animals (N=5 females). Error shading above and below the mean represents S.E.M. (Right) Group averages of the mean USV rate 1 s before laser stimulation, during a 1 s laser stimulation, and 1 s after laser stimulation (blue) compared to light-only control trials (black) and to the natural decay of vocal bouts with no stimulation (gray) for all optogenetic stimulation animals (N=5 females) and for a separate group of GFP control animals (N=5 females, green). Data for each mouse were normalized by dividing the mean USV rate pre, during, and post laser stimulation by the mean USV rate for the pre-laser period (p<0.01 for differences between ChR2 group vs. control groups during laser time; two-way ANOVA with repeated measures on one factor, **p<0.01 for interaction between group and time, followed by post-hoc pairwise Tukey's HSD tests; n.s. p>0.05 for differences between groups in post-laser period). Error bars represent S.E.M. (**E**) (Left) Representative confocal image of in situ hybridization performed on Amg$_{C/M-PAG}$ neurons (labeled with GFP by injecting AAVretro-Cre in PAG and AAV-flex-GFP in amygdala, shown in green), showing overlap with expression of VGAT (*Slc32a1*; magenta) and VGLUT2 (*Slc17a6*; blue). (Right) Quantification of overlap between GFP-labeled Amg$_{C/M-PAG}$ neurons and VGAT and VGLUT2 (N=167 from 2 hemispheres from 2 female mice: 99 cells from five 80 µm coronal sections from one mouse; 68 cells from seven 80 µm coronal sections from the other).

The online version of this article includes the following video and source data for figure 2:

**Source data 1.** Source data for *Figure 2*.

**Figure 2—video 1.** Optogenetic activation of Amg$_{C/M-PAG}$ neurons transiently suppresses ultrasonic vocalizations (USVs) in a female mouse.
https://elifesciences.org/articles/85547/figures#fig2video1

offset, the USV rate in the Amg$_{C/M-PAG}$-ChR2 female mice rebounded to a level comparable to that in all control conditions (p>0.05 for differences between groups in post-laser period; *Figure 2C–D*). When we compared these vocal suppression results to those reported in *Michael et al., 2020*, we found that vocal suppression by stimulating Amg$_{c/m-PAG}$ neurons in females was comparable to the previously reported vocal suppression by stimulating Amg$_{c/m-PAG}$ neurons in males (*Michael et al., 2020*; *Figure 3H*).

In male mice, USV-suppressing Amg$_{C/M-PAG}$ neurons are predominantly GABAergic (*Michael et al., 2020*). We used in situ hybridization to establish that 100% of Amg$_{C/M-PAG}$ cells in female mice express VGAT (*Slc32a1*), suggesting that they mediate USV suppression by directly inhibiting PAG-USV neurons (N=167/167 neurons in 2 female mice were VGAT$^+$ (*Slc32a1*), and 0/167 were VGLUT2$^+$ (*Slc17a6*); *Figure 2E*), as described in males (*Michael et al., 2020*). In summary, Amg$_{C/M-PAG}$ neurons are GABAergic and function to suppress vocalizations in female mice during social encounters with other females, similar to the manner in which these neurons suppress courtship USVs in male mice. Therefore, although female and male mice produce USVs in different social and sexual contexts, Amg$_{C/M-PAG}$ neurons in both sexes function similarly to suppress USV production.

## Amg$_{C/M-PAG}$ neurons that suppress vocalization express estrogen receptor alpha (Esr1)

Neurons that express estrogen receptor alpha (Esr1) have been implicated in a wide range of sexual and social behaviors including aggression, social recognition, courtship, and vocalization (*Chen et al., 2021*; *Hashikawa et al., 2017*; *Kudwa et al., 2006*; *Lee et al., 2014*; *Michael et al., 2020*; *Moran et al., 2020*). Indeed, several recent studies showed that optogenetically activating Esr1$^+$ neurons in the POA was sufficient to elicit long-lasting bouts of USVs in both male and female mice (*Chen et al., 2021*; *Karigo et al., 2021*; *Michael et al., 2020*). Given these observations, we wondered whether Esr1 might also be expressed in Amg$_{C/M-PAG}$ neurons. Consistent with this idea, in situ hybridization data from the Allen Brain Atlas shows expression of Esr1 mRNA in the same region of the amygdala as Amg$_{C/M-PAG}$ neurons are found in male mice (*Figure 3A*). We therefore tested the hypothesis that Amg$_{C/M-PAG}$ neurons express Esr1 using immunofluorescent Esr1 protein antibody staining. We found that nearly half of GFP-labeled Amg$_{C/M-PAG}$ neurons express Esr1 (*Figure 3B*,~49%, N=913 cells from 7 mice, 3 males and 4 females; Amg$_{C/M-PAG}$ neurons labeled by injecting PAG with AAV-retro-Cre and amygdala with AAV-flex-GFP, similar to the strategy shown in *Figure 2B*). By injecting the amygdala of Esr1-cre mice with flex-tdTomato we further found that Esr1$^+$ amygdala cells send axonal projections to the region of caudolateral PAG, the region that contains PAG-USV neurons, and consistent with our observation that a large subset of Amg$_{C/M-PAG}$ neurons are Esr1$^+$ (*Figure 3C*).

To test whether activation of Esr1$^+$ Amg$_{C/M}$ cells is sufficient to suppress USV production, we injected the Amg$_{C/M}$ of Esr1-Cre male and female mice with a Cre-dependent AAV driving ChR2 expression (*Figure 3D*). All mice were single-housed post-surgery. Optogenetic activation of Esr1$^+$ Amg$_{C/M}$ cells immediately and reversibly suppressed USV production (N=9 mice; 5 males and 4 females; *Figure 3E–G*). As with stimulation of Amg$_{C/M-PAG}$ cells, this suppression was specific to the time period of optogenetic stimulation and did not occur in light-only control conditions or in control conditions in which USV bouts were allowed to decay naturally (*Figure 3G*; *Figure 3—video 1*; p<0.001 for differences between optogenetic condition and both control conditions during the laser stimulation period, two-way ANOVA followed by post-hoc pairwise Tukey's HSD tests). Moreover, stimulating either Esr1$^+$ Amg$_{C/M}$ neurons or Amg$_{C/M-PAG}$ neurons reduced the USV rate (normalized to pre-laser baseline) to a comparable level (*Figure 3H*). Lastly, stimulating Esr1$^+$ Amg$_{C/M}$ neurons was similarly efficient for suppressing USVs in both males and females. Taken together, this evidence suggests that Esr1$^+$ Amg$_{C/M-PAG}$ neurons mediate vocal suppression in both male and female mice. Though GABAergic Esr1$^+$ neurons in the hypothalamus and glutamatergic Esr1 +neurons in the posterior amygdala have been implicated in mediating sexual behaviors and aggression, to our knowledge this is a novel role for GABAergic Esr1$^+$ neurons in the Amg$_{C/M-PAG}$.

## Analyzing circuit mechanisms that balance vocal-promoting and suppressing signals in male mice

The experiments described in the prior sections establish that activating Amg$_{PAG}$ neurons in male and female mice suppresses spontaneous USV production during social encounters with novel females,

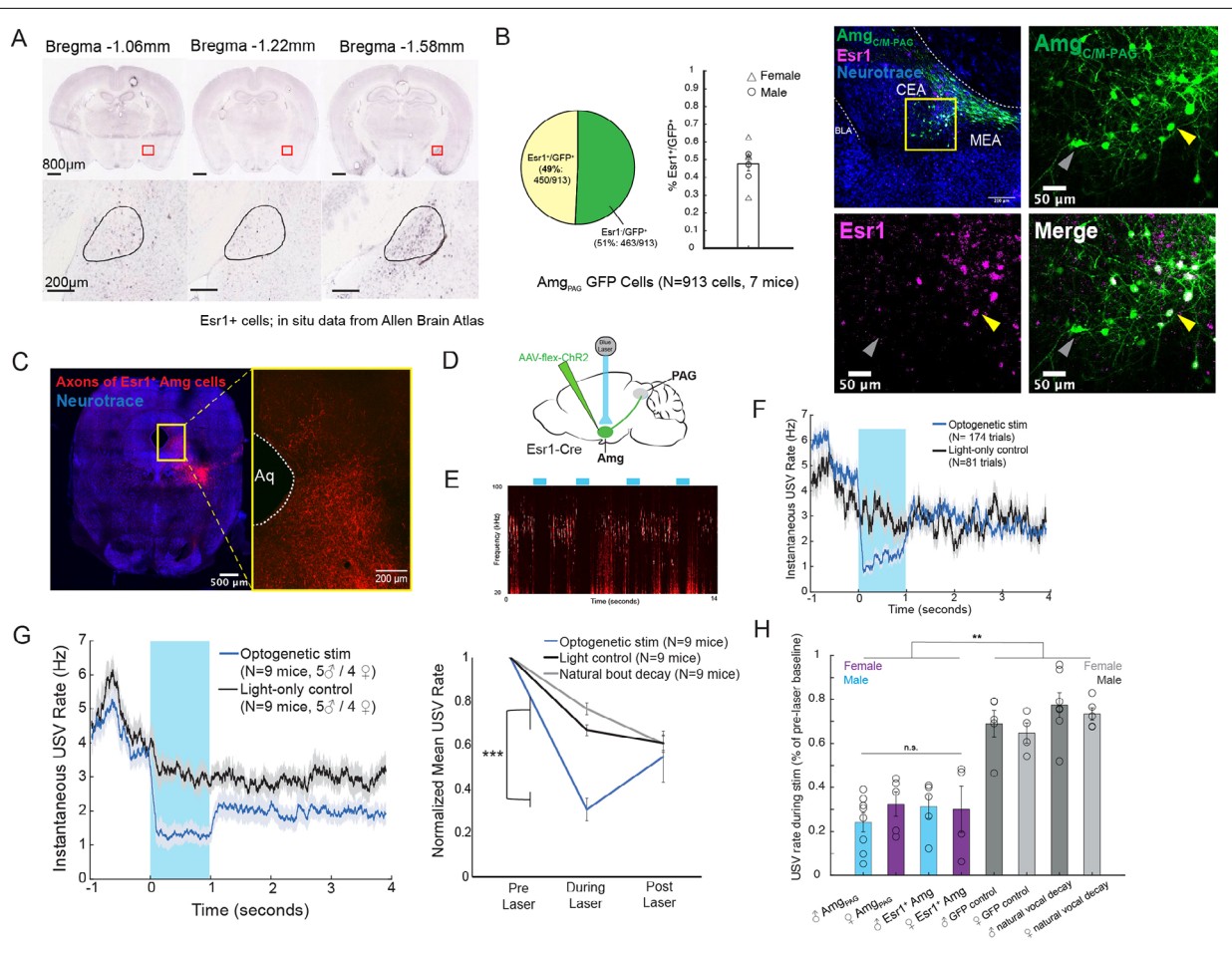

**Figure 3.** Activation of Esr1 +AmgC/M neurons suppresses vocalizations. (**A**) In situ hybridization data for Esr1 mRNA in wild-type male mouse. (Allen Institute for Brain Science. Allen Mouse Brain Atlas, available from: http://mouse.brain-map.org/experiment/show/79591677). (**B**) (Left) Quantification showing overlap of GFP-labeled Amg$_{C/M-PAG}$ neurons with Esr1 (N=913 cells from 7 hemispheres from 7 mice, on average seven 80 µm coronal sections or 130 cells per animal; 3 males and 4 females). Pie chart shows 450/913 GFP-labeled cells (49%) are Esr1$^+$. Bar graph shows distribution of %Esr1$^+$/GFP$^+$ overlap for the seven mice examined (mean = 47.73%, with S.E. of 4.08%). (Right) Representative confocal image of immunofluorescent Esr1 protein antibody staining performed on Amg$_{C/M-PAG}$ neurons (labeled with GFP, shown in green), showing overlap with expression of Esr1 (magenta). Blue is NeuroTrace. Gray arrow points to an example Esr1$^-$ Amg$_{C/M-PAG}$ neuron; yellow arrow points to an example Esr1$^+$ Amg$_{C/M-PAG}$ neuron. (**C**) Axonal projections of Esr1$^+$ amygdala cells (labeled by tdTomato) in caudolateral PAG, the region that contains PAG-USV neurons. (**D**) Viral strategy used to express ChR2 in Esr1+ Amg$_{C/M}$ neurons. (**E**) Example spectrogram showing a representative set of trials in which activation of Amg$_{C/M-PAG}$ neurons suppresses ongoing USV production. (**F**) Example traces of the USV rate during optogenetic stimulation vs a light-control condition for all trials (N=174 optogenetic trials and 62 light-only control trials) for a single representative Esr1-Cre-ChR2 animal. Error shading above and below the mean represents S.E.M. (**G**) (Left) Average USV rate traces during optogenetic stimulation and a light-only control for all Esr1-Cre-ChR2 animals (N=9 mice, 5 males and 4 females). Error shading above and below the mean represents S.E.M. (Right) Group averages of the mean USV rate 1 s before laser stimulation, during a 1 s laser stimulation, and 1 s after laser stimulation (blue) compared to light-only control trials (black) and to the natural decay of vocal bouts with no stimulation (gray) for all optogenetic stimulation animals (N=9 mice, 5 males and 4 females). Data for each mouse were normalized by dividing the mean USV rate pre, during, and post laser stimulation by the mean USV rate for the pre-laser period (***p<0.001 for differences between optogenetic condition and both control conditions during the laser stimulation period, two-way ANOVA followed by post-hoc pairwise Tukey's HSD tests). Error bars represent S.E.M. (**H**) Summary plots show USV rate normalized to pre-laser baseline during optogenetic stimulation (purple: females; blue: males) when stimulating (1) Amg$_{c/m-PAG}$ neurons in males (data were from ***Michael et al., 2020***; ***Figure 4***) vs. (2) Amg$_{c/m-PAG}$ neurons in females (***Figure 2***) vs. (3) Esr1$^+$ Amg$_{c/m}$ neurons in males and (4) Esr1$^+$ Amg$_{c/m}$ neurons in females (Figure 3). Gray bars include two controls: GFP controls and natural vocal decay controls (light gray: females; dark gray: males) (F=17.37, df = 7, p=1.105e-09, one-way ANOVA between all groups, with post-hoc Tukey's HSD tests showing that each experimental condition was significantly different from control conditions at **p<0.01, while no significant difference was found between any two experimental conditions: p$_{1,2}$=0.9538, p$_{1,3}$=0.9801, p$_{1,4}$=0.9954).

The online version of this article includes the following video and source data for figure 3:

**Source data 1.** Source data for ***Figure 3***.

*Figure 3 continued on next page*

*Figure 3 continued*

**Figure 3—video 1.** Optogenetic activation of Esr1$^+$ AmgC$_{C/M}$ neurons transiently suppresses ultrasonic vocalizations (USVs).
https://elifesciences.org/articles/85547/figures#fig3video1

that these neurons respond similarly to an aversive predator odorant that also suppresses USV production, and that they display similar molecular phenotypes. We next sought to understand how these vocal-suppressing Amg$_{C/M-PAG}$ neurons interact with vocal promoting signals to influence USV production in the service of social affiliation. Although the drive and affiliative function are well understood for male courtship USVs, the drivers and functions for USVs produced by previously isolated female mice during encounters with novel females are not known. During courtship, male mice produce vocalizations robustly (*Chabout et al., 2015*; *Neunuebel et al., 2015*; *Portfors, 2007*; *Tschida et al., 2019*; *Warren et al., 2018*). As females prefer more vocal males (*Tschida et al., 2019*), vocalizing in the presence of a female facilitates a male's mating success. In contrast, it is unclear whether the USVs that a previously isolated female produces in response to a novel female function in an affiliative or aggressive manner. For these reasons, we next sought to characterize the interaction of vocal promoting and vocal suppressing signals more fully in the amygdala of male mice.

## Characterizing monosynaptic inputs to Amg$_{C/M-PAG}$ neurons in male mice

The decision to suppress ongoing vocalizations depends on the integration of external cues with interoceptive information. Given that Amg$_{C/M-PAG}$ neurons suppress vocalizations by directly inhibiting vocal gating neurons in the PAG, they are likely to integrate a wide range of inputs that help regulate this decision. We conducted a series of additional anatomical and functional experiments in male mice to better understand the nature of this integrative process. We first conducted monosynaptically restricted transsynaptic rabies tracing from Amg$_{C/M-PAG}$ neurons to map their inputs (*Figure 4A*). To specifically label inputs to Amg$_{C/M}$ neurons that project to PAG we injected the caudolateral PAG with AAV-retro-Cre and the Amg$_{C/M}$ with Cre-dependent helper viruses that express rabies glycoprotein (G), a protein required for transsynaptic spread (*Callaway and Luo, 2015*). Two weeks later, we injected the same amygdalar regions with a pseudotyped G-deleted rabies virus, which when complemented by the wild-type rabies G glycoprotein, resulted in expression of GFP in neurons that make synapses onto Amg$_{C/M-PAG}$ neurons (*Figure 4A*; N=4 males).

This approach revealed that Amg$_{C/M-PAG}$ neurons in male mice receive monosynaptic input from numerous cortical regions that process a wide variety of sensory and interoceptive information, including the piriform, motor, somatosensory, insular and auditory cortices (*Figure 4B*; *Bhattacharjee et al., 2021*; *Gogolla, 2017*; *Shipley and Ennis, 1996*; *Zatorre et al., 2002*). We also observed that Amg$_{C/M-PAG}$ neurons receive input from many other forebrain and midbrain regions, including the medial and lateral POA, the bed nucleus of the stria terminalis (BNST), the ventral tegmental area, and the nucleus accumbens shell (*Figure 4C–D*), which have been variously implicated in defensive behaviors, sexual-social interactions, reward and satisfaction (*Breitfeld et al., 2015*; *Michael et al., 2020*; *Paredes, 2003*; *Salgado and Kaplitt, 2015*).

Of particular interest to us was the labeling of cell bodies in the medial POA (*Figure 4E*), a region that contains neurons whose activation elicits USVs via a disinhibitory circuit motif within the PAG (*Chen et al., 2021*; *Michael et al., 2020*). This anatomical organization raises the possibility that Amg$_{C/M-PAG}$ cells are a cellular locus where USV-suppressing information can be weighed against USV-promoting signals from the POA. Therefore, we focused on characterizing the social and USV-related information that POA axons convey to the Amg$_{C/M}$, and the function of the synapses that POA neurons make on Amg$_{C/M-PAG}$ neurons in male mice.

## In male mice, POA inputs to the Amg$_{C/M}$ are active during affiliative social encounters and USV production

To measure the activity of POA inputs to the Amg$_{C/M}$, we used intersectional methods to express GCaMP8s in Esr1$^+$ POA neurons and recorded bulk fluorescence from their axon terminals in the Amg$_{C/M}$ (*Figure 5A*; N=5 male Esr1-Cre mice). We then measured the activity of POA axons in the Amg$_{C/M}$ during the male's social interactions with novel female and male mice, 2MT, or novel plastic toys. In contrast to the activity of Amg$_{C/M-PAG}$ neurons measured in these various conditions (*Figure 1G*),

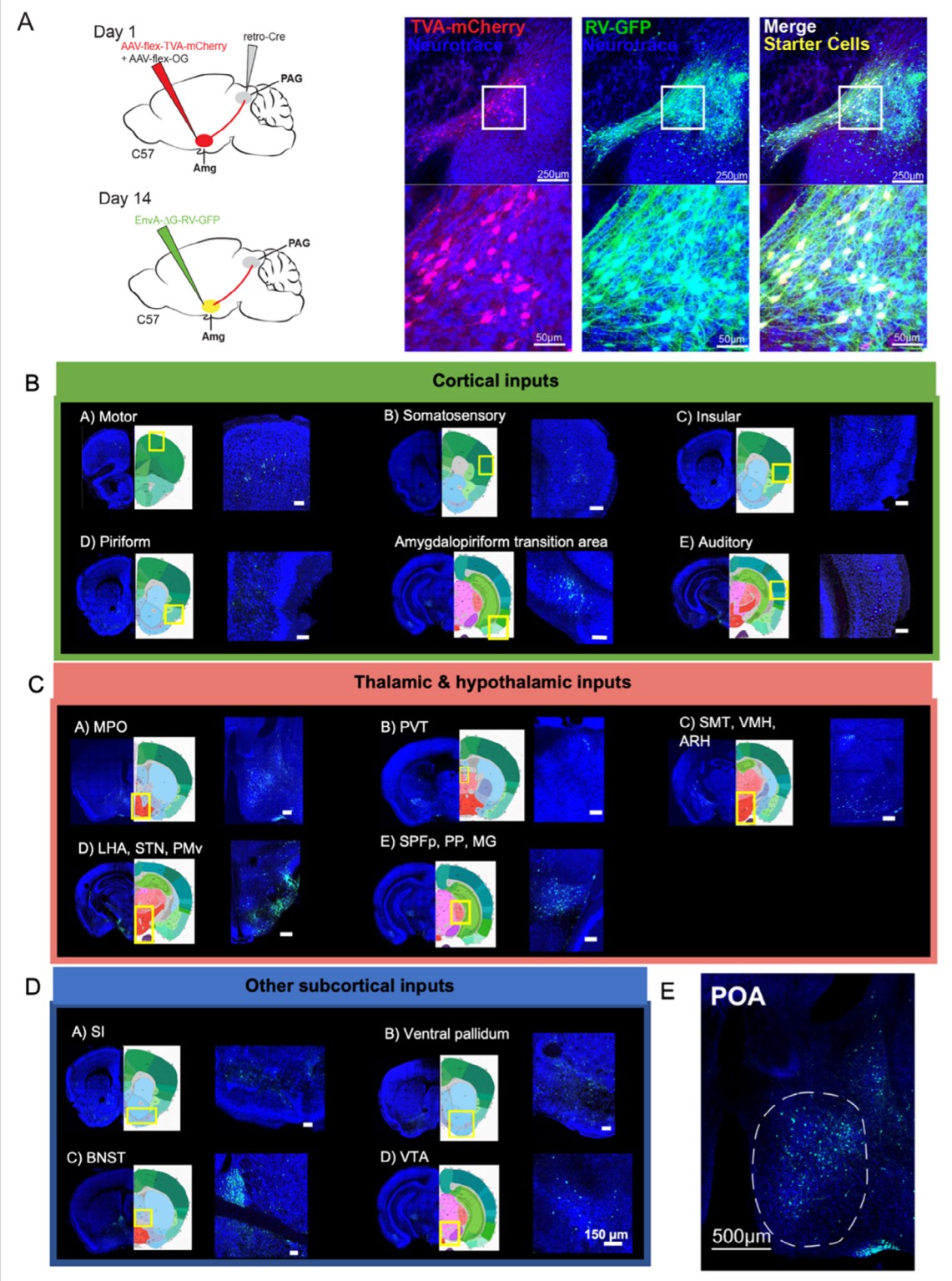

**Figure 4.** Monosynaptic inputs to Amg$_{C/M-PAG}$ neurons in male mice. (**A**) (Left) Viral strategy shown for transsynaptic labeling of direct inputs to Amg$_{C/M-PAG}$ neurons (performed in N=4 males). (Right) Confocal images are shown of starter Amg$_{C/M-PAG}$ neurons. (**B–E**) Confocal images are shown of upstream neurons labeled in the cortical and subcortical areas, including the preoptic area of the hypothalamus (POA). Scale bars in B, C, D are 150 µm.

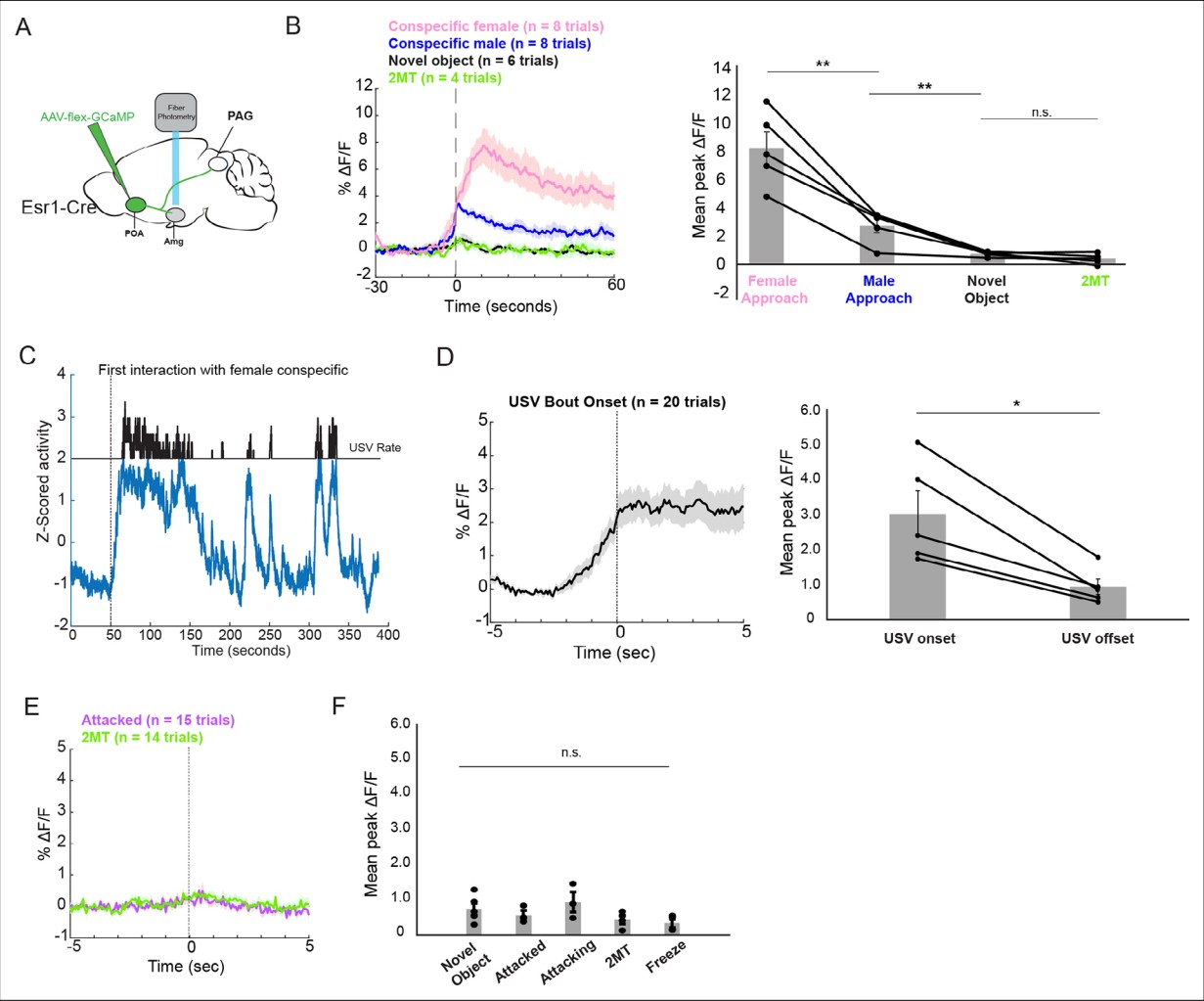

**Figure 5.** POA inputs to the Amg$_{C/M}$ are active during affiliative social encounters and USV production in male mice. (**A**) Viral strategy for photometry recording of Esr1$^+$ POA terminal activity in Amg$_{C/M}$ region (POA$_{Esr1-Amg}$ terminal activity). (**B**) Average POA$_{Esr1-Amg}$ activity in an example male mouse (left) and mean peak POA$_{Esr1-Amg}$ activity from 5 males (right) at the onset of approaching female mice, at the onset of non-aggressive social interactions with other male mice, and at the onset of or during trials where the mouse encountered a dish containing 2MT or a novel plastic toy (N=5 males; **p<0.01; n.s. p>0.05; one-way ANOVA followed by post-hoc pairwise Tukey's HSD tests). Error bars represent S.E.M. (**C**) POA$_{Esr1-Amg}$ terminal activity as a male vocalized to a female. (**D**) Average POA$_{Esr1-Amg}$ terminal activity in an example male mouse (left) and mean peak POA$_{Esr1-Amg}$ activity from 5 males (right) at USV bout onset (N=5 males; n.s. p<0.05, paired t test). (**E**) Average POA$_{Esr1-Amg}$ terminal activity in an example male mouse during aggressive interactions with other males and investigation of a dish containing 2MT. (**F**) Mean peak POA$_{Esr1-Amg}$ activity from 5 males during aggressive interactions with other males, investigation of novel object or a dish containing 2MT, or during freezing in the presence of 2MT (n.s. p>0.05; one-way ANOVA test).

The online version of this article includes the following source data for figure 5:

**Source data 1.** Source data for *Figure 5*.

POA$_{Esr1-Amg}$ terminal activity showed a large and sustained increase at the onset of social interactions with female mice, a moderate sustained increase at the onset of non-aggressive social interactions with other male mice, and no response at the onset of or during trials where the mouse encountered a dish containing 2MT or a novel plastic toy (*Figure 5B*). In addition to the large and sustained increase in activity upon the introduction of a social partner, POA$_{Esr1-Amg}$ terminal activity further increased at the onset of subsequent social interactions and at the onset of individual USV bouts (*Figure 5C–D*). However, POA$_{Esr1-Amg}$ terminal activity remained low when aligned to the onsets of the mouse's investigation of a dish containing 2MT or the onsets of attacks by male conspecifics (*Figure 5E and F*). Taken together, these findings support the idea that POA$_{Esr1-Amg}$ axons in the Amg$_{C/M-PAG}$ of male mice are highly active during social encounters in which USVs are produced, but not in situations in which Amg$_{C/M-PAG}$ activity is high and USV production is suppressed. Therefore, in addition to being strongly

activated by stimuli or aggressive social interactions that suppress USV production (*Figure 1D, F and J*), Amg$_{C/M-PAG}$ neurons receive inputs from the POA that convey information about pro-vocal social contexts and USV production.

## In male mice, POA neurons that provide input to the Amg$_{C/M}$ are Esr1$^+$ and optogenetically activating them is sufficient to promote USV production

These various observations strongly suggest that the POA neurons that provide input to the amygdala are actually USV-promoting neurons. Because activating POA$_{PAG}$ cells that express Esr1 is sufficient to promote USV production (*Chen et al., 2021*; *Karigo et al., 2021*; *Michael et al., 2020*), we first investigated whether POA cells that project to the Amg$_{C/M}$ (POA$_{AMG}$ cells, labeled by intersectional methods; *Figure 6A*) are part of this Esr1-expressing subpopulation of POA cells. We found that the majority (~82%) of GFP-labeled POA$_{AMG}$ cells express Esr1 (*Figure 6B*, N=298/362 cells from 2 male mice). We further noted that axon terminals from GFP-labeled POA$_{AMG}$ cells were present in the caudolateral PAG, the same region where PAG-USV neurons are located (*Figure 6C*, see also *Tschida et al., 2019*). To directly test whether POA$_{AMG}$ cells are USV-promoting, we used intersectional viral methods to express ChR2 in POA$_{AMG}$ cells (*Figure 6D*). We found that exciting the cell bodies of these ChR2-expressing POA$_{AMG}$ cells with blue light was sufficient to trigger USVs in male mice in the absence of social cues (*Figure 6E–G*; *Figure 6—video 1*; N=4 male mice showed increased USV rates in response to this optogenetic stimulation). The effects on mean USV rate, success rate of eliciting USVs, and the mean latency to elicited USVs using the POA$_{AMG}$ optogenetic stimulation approach were not significantly different from the USV-promoting effects achieved by stimulating POA$_{PAG}$ neurons or Esr1$^+$ POA terminals in PAG (*Michael et al., 2020*; *Figure 6H*). Therefore, the POA cells that convey pro-vocal social signals and USV-related signals to the Amg$_{C/M}$ are Esr1$^+$ also project to the caudolateral PAG and directly promote USV production.

## Vocalization-triggering POA$_{PAG}$ cells make inhibitory synapses onto Amg$_{C/M-PAG}$ neurons

Prior studies show that the vast majority of POA$_{PAG}$ neurons, including those that promote USV production, release the inhibitory neurotransmitter GABA (*Chen et al., 2021*; *Michael et al., 2020*). Therefore, our current observations raise the interesting possibility that the Amg$_{C/M-PAG}$ cells that suppress USVs are *directly* inhibited by the POA$_{PAG}$ neurons that promote USV production. To test this hypothesis, we first injected the PAG with AAV-retro-Cre, the POA with AAV-flex-ChR2, and the amygdala with AAV-flex-tdTomato (*Figure 6I*). Several weeks later, we performed in vitro whole-cell voltage clamp recordings from visually identified Amg$_{C/M-PAG}$ neurons while optogenetically activating POA$_{PAG}$ axon terminals in brain slices containing the amygdala. We visually targeted our recordings to tdTomato$^+$ Amg$_{C/M-PAG}$ neurons in the presence of TTX and 4AP to isolate monosynaptic currents. Optogenetically activating POA$_{PAG}$ axon terminals in the amygdala evoked inhibitory postsynaptic currents (IPSCs) in a majority (9/13) of Amg$_{C/M-PAG}$ (td-Tomato+) neurons (*Figure 6J*; mean current = 141.9 pA at 0 mV in TTX/4AP; currents with reversal potentials at –70 mV were identified as IPSCs, and no EPSCs were observed at –70 mV in the presence of TTX and 4AP). Bath application of the GABA$_A$ receptor antagonist gabazine SR-95531 abolished these optogenetically evoked IPSCs in all four cases in which it was applied (*Figure 6J*, 4/9 recordings). Therefore, POA$_{PAG}$ cells provide monosynaptic inhibitory input onto Amg$_{C/M-PAG}$ neurons. These various findings support the idea that USV-suppressing Amg$_{C/M-PAG}$ neurons, which are strongly activated by predator cues and social threats, are also directly inhibited by USV-promoting POA neurons.

## Discussion

Aversive stimuli and affiliative social stimuli both exert strong influence on an animal's decision to vocalize, but in opposite directions. How vocal-suppressing information is encoded, and how this information interacts with vocal-promoting information in the brain has been unclear. Here, we provide evidence in mice that vocal-suppressing neurons in the amygdala (Amg$_{C/M-PAG}$ neurons) encode information about predators and threatening rivals, and that they influence the decision to vocalize by integrating this information with that about potential social partners and mates.

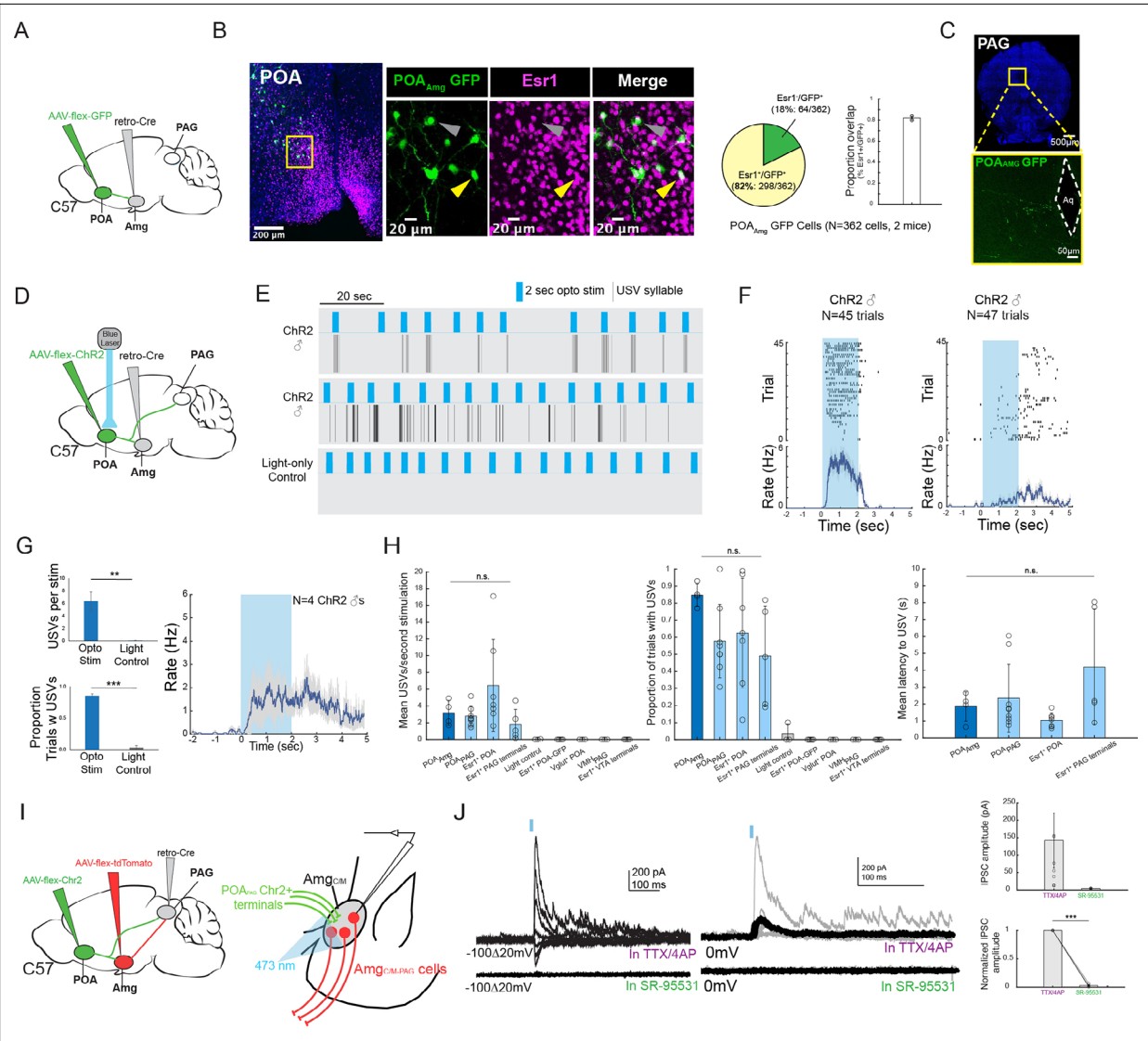

**Figure 6.** POA$_{Amg}$ cells are Esr1$^+$, project to the PAG, promote USV production, and make inhibitory synapses onto Amg$_{C/M-PAG}$ neurons. (**A**) Viral strategy for labeling Amg$_{C/M}$-projecting POA neurons. (**B**) (Left) Confocal images showing GFP-labelled Amg$_{C/M}$-projecting POA neurons and Esr1 staining. Gray arrow points to an example Esr1$^-$ POA$_{Amg}$ neuron; yellow arrow points to an example Esr1$^+$ POA$_{Amg}$ neuron. (Right) Quantification of Esr1$^+$ POA$_{Amg}$ neurons (N=362 cells from 2 hemispheres from 2 mice: 173 cells from 6 80 μm coronal sections from one mouse; 189 cells from 6 80 μm coronal sections from the other mouse). Pie chart shows 298/362 GFP-labeled cells are Esr1$^+$ (82.32%). Bar graph shows distribution of %Esr1$^+$/GFP$^+$ overlap in the two mice examined (mean = 82.32% with S.E. of 1.89%). (**C**) Axon terminals of GFP-labelled Amg$_{C/M}$-projecting POA neurons in PAG. (**D**) Viral strategy for optogenetic activation of Amg$_{C/M}$-projecting POA neurons in male mice (N=4). (**E**) Top two plots show USV syllables (black line) elicited in two representative mice following 2s-long, 10 Hz optogenetic activation (blue) of ChR2-expressing POA$_{AMG}$ cells. Stimulation was performed on an animal alone in his home cage. Bottom plot shows an absence of USVs during light control, where the 2s-long, 10 Hz laser (blue) shined above the mouse's head but was not connected to the ferrule. (**F**) Raster plots show USVs elicited from all trials in the two representative mice shown in (**E**). (**G**) (Left) Bar plots compare the mean number of USVs elicited per stimulation (Top), and the success rate of eliciting USVs (**p<0.01; paired *t* test) (Bottom) during optogenetic stimulation and light control for N=4 male mice (***p<0.001; paired *t* test). (Right) Mean USV rate plotted for N=4 males following optogenetic stimulation of ChR2-expressing POA$_{AMG}$ cells in the absence of social cues (alone in home cage). Gray shading above and below the mean represents S.E.M. (**H**) Summary plots show comparison among four experimental approaches for eliciting USVs. Vocalizations elicited by optogenetically stimulating POA$_{Amg}$ neurons (current study; dark blue) did not differ significantly from those elicited by optogenetically stimulating (1) POA$_{PAG}$ neurons, (2) Esr1$^+$ POA neurons or (3) Esr1$^+$ POA terminals at PAG in terms of mean USV rate (Kruskal-Wallis test, Chi-sq=6.36, df = 3, p=0.0954), success rate (Kruskal-Wallis test, Chi-sq=5.1, df = 3, p=0.1645), and mean latency (Kruskal-Wallis test, Chi-sq=7.12, df = 3, p=0.0681). The four experimental approaches include POA$_{AMG}$ neurons (dark blue, N=4 males from current study), POA$_{PAG}$ neurons (light blue, N=9 males from *Michael et al., 2020*), Esr1$^+$POA neurons (light blue, N=7 males from *Michael et al., 2020*), Esr1$^+$ POA axon terminals within the PAG (light blue, N=5 males from *Michael et al., 2020*). For left and middle panels, the following controls were also shown for reference: light control (N=4 males from current study), GFP-

*Figure 6 continued on next page*

*Figure 6 continued*

expressing Esr1+ POA neurons (N=5 males from *Michael et al., 2020*), VGLUT2+ POA neurons (N=3 males from *Michael et al., 2020*), VMH$_{PAG}$ neurons (N=3 males from *Michael et al., 2020*), and Esr1+ POA axon terminals within the ventral tegmental area (VTA) (N=4 males from *Michael et al., 2020*). Error bars show S.D. (**I**) Viral strategy for in vitro whole-cell voltage clamp recordings from visually identified Amg$_{C/M-PAG}$ neurons while optogenetically activating POA$_{PAG}$ axon terminals in brain slices containing the amygdala. (**J**) (Left) Light-evoked IPSCs recorded in TTX/4AP (observed in N=9 of 13 td-Tomato-tagged Amg$_{C/M-PAG}$ neurons from N=5 males) were abolished by application of gabazine (N=4 cells also recorded in gabazine). IPSC amplitude refers to the peak of the light-evoked current at 0 mV holding potential. (Right) (Top) Mean IPSC amplitude recorded in TTX/4AP (N=9; one cell with an IPSC amplitude 757.79 pA not shown on the plot) and SR-95531 (N=4). (Bottom) Comparison of IPSC amplitude for the four cells recorded in both TTX/4AP and SR-95531, normalized to IPSC amplitude in TTX/4AP (***p=6.32e-06; paired *t* test,). Error bars represent S.E.M.

The online version of this article includes the following video, source data, and figure supplement(s) for figure 6:

**Source data 1.** Source data for *Figure 6*.

**Figure supplement 1.** Cannula placement and ChR2 expression in POA$_{Amg}$ neurons.

**Figure 6—video 1.** Optogenetic activation of POA$_{Amg}$ neurons elicits ultrasonic vocalizations (USVs).
https://elifesciences.org/articles/85547/figures#fig6video1

Using behavioral methods, we established that 2MT, a potent analog of the fox urine odorant TMT, suppresses USVs produced by male and female mice. Moreover, using fiber photometric calcium imaging, we found that PAG-projecting Amg$_{C/M}$ neurons (Amg$_{C/M-PAG}$ neurons) in both females and males were strongly activated during exposure to 2MT but not when the animal subsequently exhibited 2MT-evoked freezing or was exposed to other neutral odorants. We also observed that Amg$_{C/M-PAG}$ neurons were excited during aggressive social encounters between male mice, particularly when a male was under attack, a context when we never detected USV production. These results indicate that Amg$_{C/M-PAG}$ neurons provide a neural substrate for encoding threating social and environmental stimuli in either sex of mice. Further, using a novel social isolation paradigm (*Tschida et al., 2019*), we elicited robust USVs in female mice. We showed that stimulation of Amg$_{C/M-PAG}$ neurons not only suppresses USV production in male mice in the context of courtship (*Michael et al., 2020*), but also operates similarly to suppress a form of social USV production in pairs of interacting female mice. Thus, although male and female mice produce USVs in different amounts and in different social and reproductive contexts, the amygdala >PAG pathway encodes information about interaction with threatening stimuli, and it functions similarly to effectively suppress vocalizations across contexts and in both sexes.

We also established that these USV-suppressing Amg$_{C/M-PAG}$ neurons are predominantly GABAergic, with a subset expressing Esr1 in both female and male mice. While previous studies have shown that activating Esr1+ neurons in the POA promotes USVs (*Chen et al., 2021*; *Michael et al., 2020*), we found that activating Esr1+ neurons in the Amg$_{C/M}$ region robustly suppresses USVs in both male and female mice. This result suggests that Esr1 may serve as a more widespread marker of the neural circuits that regulate USV production across sexes. It also adds to the existing literature in underscoring the diverse, and sometimes opposing, social functions of Esr1 + cells (*Chen et al., 2021*; *Hashikawa et al., 2017*; *Kudwa et al., 2006*; *Lee et al., 2014*; *Michael et al., 2020*; *Moran et al., 2020*). Together, our results indicate a similar functional and anatomical organization of vocal suppressing pathways in both sexes and provide insight into how certain aversive stimuli act centrally in the mouse's brain to suppress vocal communication.

Beyond establishing a common function of Amg$_{C/M-PAG}$ neurons in both male and female mice, the current study also provides a deeper characterization of how Amg$_{C/M-PAG}$ neurons integrate vocal promoting and vocal suppressing signals in male mice. The rationale for this male-centric focus is that the motivational drive and affiliative social functions of USV production in male mice are well understood, whereas the motivational drive and social functions of female vocalizations remain elusive. In particular, it is unclear whether the USVs that previously isolated females produce in response to a novel female, as described by *Zhao et al., 2021*, function to promote affiliation or to signal aggression.

In adult male mice, robust vocal production to females facilitates mating success, as females prefer more vocal males (*Tschida et al., 2019*). Yet, in the wild, sexual and social stimuli may coincide with aversive stimuli. For instance, a male may encounter a potential mate in an area where he can also detect potential threats, such as the smell of a predator. In this situation, how does the male balance the drive to reproduce with the drive to survive? In our behavioral experiments, we observed that while mice vocalized significantly less in the presence of both predator odor and a female social

partner, they still vocalized. This observation provides naturalistic evidence that there exists a logic where pro-social drive can eclipse the drive to suppress vocalizations.

To elucidate the interaction of vocal-suppressing and vocal promoting signals in males, we used monosynaptic rabies virus tracing to map an extensive array of cortical and subcortical neurons that provide input onto these $Amg_{C/M-PAG}$ neurons. These include GABAergic neurons in the medial preoptic area that we and others previously demonstrated promote USV production by disinhibiting PAG-USV neurons (*Chen et al., 2021*; *Michael et al., 2020*). Notably, we showed that a subset of GABAergic POA neurons that project to the PAG ($POA_{PAG}$ neurons) also make inhibitory synapses on $Amg_{C/M-PAG}$ neurons. These $POA_{PAG}$ inputs to $Amg_{C/M-PAG}$ neurons are activated in USV-promoting and USV-producing contexts, and optogenetically activating POA neurons that provide inputs to the $Amg_{C/M}$ region was sufficient to promote USV production in socially isolated male mice. Consequently, in addition to PAG-USV neurons, our study advances $Amg_{C/M-PAG}$ neurons as sites where competing types of sexual, social, and environmental information are weighed to regulate the mouse's decision to vocalize (*Figure 7*).

Indeed, the PAG has long been recognized as an obligatory for vocalizations in a wide range of mammalian species (*Jürgens, 2009*; *Jürgens, 2002*; *Nieder and Mooney, 2020*), and the molecular genetic identification of USV-gating neurons in the mouse has opened the door to a more systematic analysis of the circuitry that underlies the mouse's decision to vocalize (*Michael et al., 2020*; *Tschida et al., 2019*). Our recent prior study exploited this advance to establish that PAG-USV neurons receive USV-promoting inputs from the POA and USV-suppressing inputs from $Amg_{C/M-PAG}$ neurons (*Michael et al., 2020*). In the current study, we extend these prior observations by showing that the Amg-PAG pathway responds to predator odorants and, in the male, threatening social encounters. Collectively, these studies advance the PAG as one site where vocal promoting and vocal suppressing information can be weighed to influence the decision to vocalize.

Beyond underscoring that the PAG is a site where such competing information is integrated to influence the decision to vocalize, the present study reveals that vocal-suppressing neurons in the amygdala also weigh and compare vocal promoting and vocal suppressing information. In particular, here we showed that USV-suppressing $Amg_{C/M-PAG}$ neurons are directly inhibited by USV-promoting neurons in the POA. In contrast, a prior study showed that $Amg_{C/M-PAG}$ neurons do not project back into the USV-promoting region of the POA (*Michael et al., 2020*). Thus, although $Amg_{C/M-PAG}$ neurons may alter POA activity through indirect pathways, the asymmetry of this circuit supports a logic where the reproductive and social benefits of vocalizing may outweigh the potential costs of advertising a male mouse's location to predators and aggressive social rivals. One limitation of the current study is that we did not test whether optogenetic activation of POA terminals in the $Amg_{C/M}$ region in the presence of 2MT and in the absence of a live social partner is sufficient to elicit vocalizations. While the present efforts provide insights into the vocalization decision-making process, future experiments are needed to test the full dynamic range of interactions between vocal-promoting and vocal-suppressing afferents to the amygdala and the PAG. It is also important that future studies examine whether this hierarchical asymmetry among POA-Amg-PAG mediating vocal decisions is present in females as well. In summary, the present study provides insight into how environmental and social factors are encoded and weighed at several levels of a hierarchically organized circuit to regulate social vocalizations in adult male and female mice. Given the conserved nature of many of the neural components examined here, a similar nested circuit design may play an important role in regulating vocalizations in other mammals, including humans.

## Materials and methods
### Contact for reagent and resource sharing
Further information and requests for resources and reagents should be directed to the corresponding authors, Richard Mooney (mooney@neuro.duke.edu).

### Experimental models and subject details
Animal statement
All experiments were conducted according to protocols approved by the Duke University Institutional Animal Care and Use Committee.

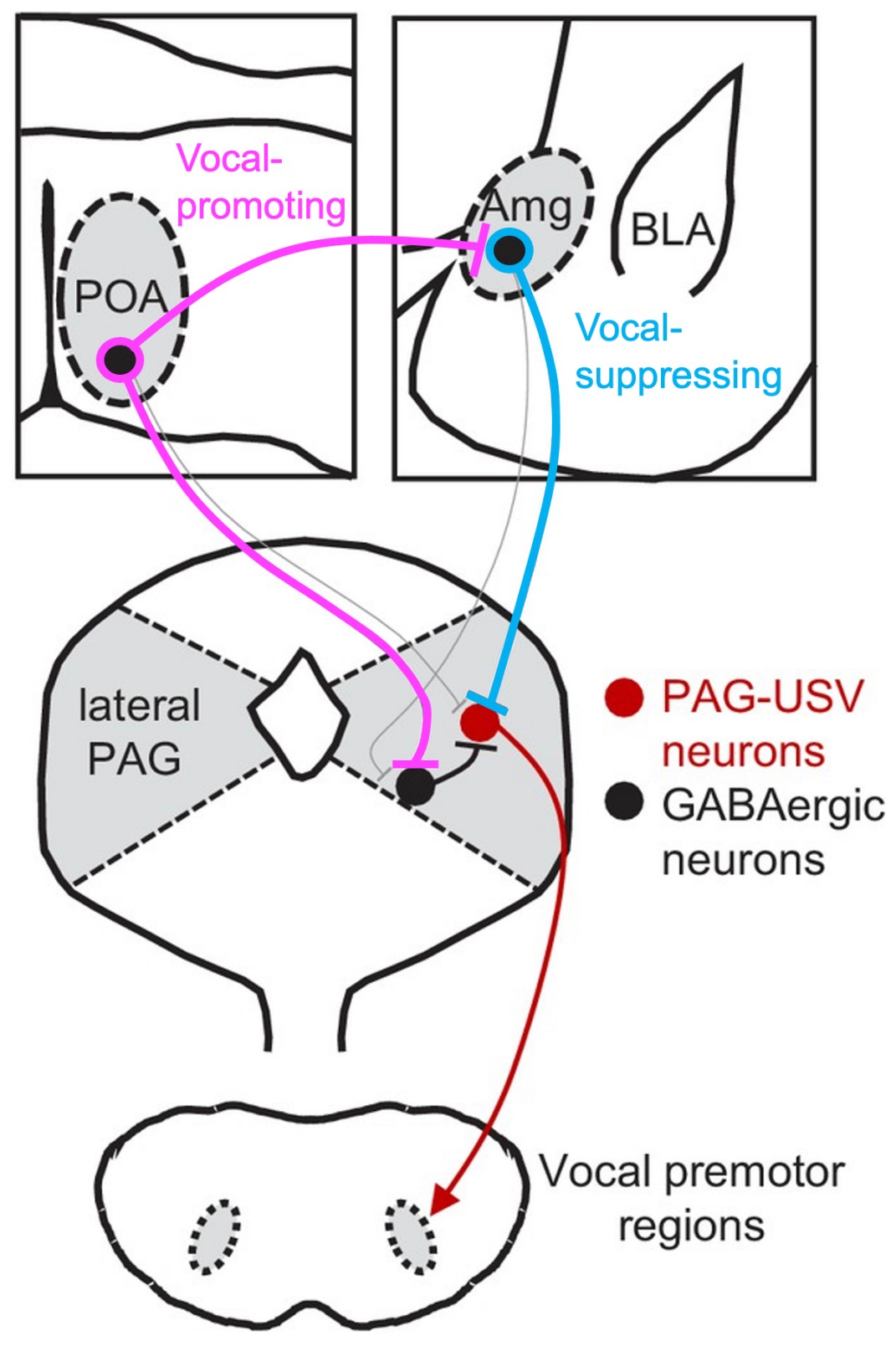

**Figure 7.** Model of nested circuits that mediate the decision to vocalize. Inhibitory neurons within the $Amg_{C/M}$ encode threatening stimuli and suppress vocalizations via their direct inhibitory input to PAG-USV neurons, which gate USV production via their excitatory input to downstream vocal premotor neurons. At the same time, $Amg_{C/M-PAG}$ neurons receive direct input from inhibitory neurons in the POA, which encode social stimuli and promote

*Figure 7 continued on next page*

*Figure 7 continued*

vocalizations via their axon collaterals that disinhibit PAG-USV neurons. Together, Amg$_{C/M\text{-}PAG}$ neurons, along with POA$_{PAG}$ and PAG-USV neurons, form a nested hierarchical circuit in which environmental and social information converges to influence the decision to vocalize.

## Animals

For optogenetic activation, fiber photometry, and transsynaptic tracing experiments, the following mouse lines from Jackson labs were used: C57BL/6 (000664) and Esr1-Cre (017911). For whole-cell recording experiments C57BL/6 mice were used (000664).

## Method details

### Viruses

The following viruses and were used: AAV2/1-hSyn-flex-ChR2-eYFP (Addgene), AAV-pgk-retro-Cre (Addgene), AAV-flex-GFP (Addgene), AAV-flex-tdTomato (Addgene), AAV-flex-GCamp8s (Addgene), and AAV-flex-oG (Duke Viral Vector Core). EnvA-ΔG-RV-GFP and AAV-flex-TVA-mCherry were produced in house as previously described (*Rodriguez et al., 2017*; *Sakurai et al., 2016*; *Tschida et al., 2019*; *Michael et al., 2020*). The final injection coordinates were as follows: POA, AP = 0.14 mm, ML = 0.3 mm, DV = 5.5 mm; Amg$_{C/M}$, AP = −1.5 mm, ML = 2.3 mm, DV = 4.6 mm; PAG, AP = −4.7 mm, ML = 0.7 mm, DV = 1.75 mm. Viruses were pressure-injected with a Nanoject II (Drummond) at a rate of 4.6 nL every 15 s.

### Vocalization behavioral test

in the neutral condition, a stimulus female mouse along with a dish containing clean bedding pipetted with 10 µL of DI water was placed into the home cage of either a male or a female resident; in the 2MT condition, a stimulus female mouse along with a dish containing clean bedding pipetted with 10 µL of 2MT (Sigma Aldrich M83406; 1:50 dilution, amount and dilution same as used in *Bruzsik et al., 2021*) was placed into the home cage of either a male or a female resident. Video and USVs were recorded for 5 min after the introduction of stimulus mouse and odorant dish.

### USV recording and analysis

USVs were recorded using an ultrasonic microphone (Avisoft, CMPA/CM16), amplified (Presonus TubePreV2), and digitized at 250 kHz (Spike 7, CED). USVs were detected using codes modified from the Holy lab (http://holylab.wustl.edu/) using the following parameters (mean frequency >45 kHz; spectral purity >0.3; spectral discontinuity <0.85; min. USV duration = 5ms; minimum inter-syllable interval = 30ms).

### Fiber photometry

On the day of testing, the implanted fiber optic cannula was plugged into a three-channel multi-fiber photometry system (Neurophotometrics, Ltd.). In this system, 470 nm (for imaging in green), 560 nm (for imaging in red), and 415 nm (control signal to detect calcium-independent artifacts) LEDs are bandpass filtered and directed down a fiber optic patch cord which is coupled to the implanted fiber optic cannula. Emitted GCaMP fluorescence is captured through the patch cord and split by a 532 dichroic, bandpass filtered, and focused onto opposite sides of a CMOS camera. Data were acquired using Bonsai software by drawing a region of interest (ROI) around the green image of the patch cord and calculating the mean pixel value. The blue and green channels were median filtered and the blue channel was fit to and subtracted from the green signal. During recordings, stimuli including a dish containing either 10 µL of 2MT (Sigma Aldrich M83406; 1:50 dilution, amount and dilution same as used in *Bruzsik et al., 2021*), 10 µL of a control odor (ethyl tiglate), a novel plastic 'toy', or the soiled bedding of male or female conspecifics, or a live novel female or male intruder were introduced one at a time to the experiment mouse's home cage. Video, audio and calcium signals were recorded as the experiment mouse freely interacted with the presented stimulus. The resulting calcium signal was analyzed using custom matlab codes.

## Behavior scoring

Behaviors examined in the fiber photometry experiments (approach, non-aggressive and aggressive social interactions, freezing behaviors) were scored manually by experimenters who were blind to the mouse genotype, recording site/experiment and the photometry recordings. An approach to the dish was timestamped when the mouse extended their snout to the dish to sniff the dish content. A non-aggressive interaction refers to chemoinvestigation (or anogenital sniffing) of the intruder. An aggressive interaction refers to violent fighting, where the resident and the intruder were locked together rolling, biting, kicking, and wrestling with each other. Freezing behavior was operationally defined as when the animal remained motionless for at least 3 continuous seconds.

## In situ hybridization using hybridization chain reaction (HCR)

In situ hybridization was performed using hybridization chain reaction (HCR v3.0, Molecular Instruments) as previously described in *Michael et al., 2020*, with VGAT (*Slc32a1*) and VGLUT2 (*Slc17a6*) probes being applied to 80-µm-thick coronal floating sections including the $Amg_{C/M}$ region. Quantification of overlap between GFP-labeled $Amg_{C/M-PAG}$ neurons and VGAT and VGLUT2 were performed on all sections that contained GFP labeling, which ranged from 5 to 7 sections per mouse.

## In vivo optogenetic stimulation

For optogenetic stimulation of $Amg_{C/M-PAG}$ cells in female mice, the caudolateral PAG of C57BL/6 female mice was injected with AAV-pgk-retro-Cre (100 nL) and the $Amg_{C/M}$ was injected with AAV-flex-ChR2 (100 nL) and the optogenetic ferrule was placed above the $Amg_{C/M}$ cell bodies. For optogenetic stimulation of $Amg_{C/M}$ Esr1$^+$ neurons the $Amg_{C/M}$ of Esr1-cre male and female mice was injected with AAV-flex-ChR2 (100 nL) and the optogenetic ferrule was placed above the $Amg_{C/M}$ cell bodies. For optogenetic stimulation of $POA_{Amg}$ cells the $Amg_{C/M}$ of C57BL/6 male mice was injected with AAV-pgk-retro-Cre (100 nL) and the medial POA was injected with AAV-flex-ChR2 (100 nL) and the optogenetic ferrule was placed above the $POA_{Amg}$ cells bodies.

Commercially available (RWD) optogenetic ferrules were implanted in the same surgeries as viral injection just above target brain locations and were fixed to the skull using Metabond (Parkell). Neurons were optogenetically activated with illumination from a 473 nm laser (3–15 mW) at 10–20 Hz (50ms pulses, 2–10 s total) or with phasic laser pulses (1–2 s duration). Laser stimuli were driven by computer-controlled voltage pulses (Spike 7, CED). For stimulation of $Amg_{C/M-PAG}$ neurons, $Amg_{C/M}$ Esr1$^+$ neurons, the laser was manually triggered each time the mouse began a bout of vocalization towards a social partner that lasted several seconds. For stimulation of $POA_{Amg}$ neurons stimulation was triggered manually at regular intervals while the animal was alone.

## Post-hoc visualization of viral labeling

Mice were deeply anesthetized with isoflurane and then transcardially perfused with ice-cold 4% paraformaldehyde in 0.1 M phosphate buffer, pH 7.4 (4% PFA). Dissected brain samples were post-fixed overnight in 4% PFA at 4 °C, cryoprotected in a 30% sucrose solution in PBS at 4 °C for 48 hr, frozen in Tissue-Tek O.C.T. Compound (Sakura), and stored at –80 °C until sectioning. To visualize viral labeling post-hoc, brains were cut into 80 µm coronal sections, rinsed 3 x in PBS, and processed for 24 hr at four degrees with NeuroTrace (1:500 Invitrogen) in PBS containing 0.3% Triton-X. Tissue sections rinsed again 3×10 min. in PBS, mounted on slides, and coverslipped with Fluoromount-G (Southern Biotech). After drying, slides were imaged with a 10 x objective on a Zeiss 700 laser scanning confocal microscope.

## Esr1 immunohistochemistry

To examine the proportion of $Amg_{c/m-PAG}$ cells that express Esr1, $Amg_{c/m-PAG}$ cells were labeled by injecting AAVretro-Cre into PAG and AAV-flex-GFP in the amygdala (more details described in the 'viruses' section). Two weeks after the injection, mice were deeply anaesthetized with isoflurane and then transcardially perfused with ice-cold 4% paraformaldehyde in 0.1 M phosphate buffer, pH 7.4 (4% PFA). Dissected brain samples were post-fixed overnight in 4% PFA at 4 °C, cryoprotected in a 30% sucrose solution in PBS at 4 °C for 48 hr, frozen in Tissue-Tek O.C.T. Compound (Sakura), and stored at –80 °C until sectioning. Brains were cut into 80 µm coronal sections, rinsed 3 X in PBS, permeabilized

for 3 hr in PBS containing 1% Triton X (PBST), and then blocked in 0.3% PBST containing 10% Blocking One (Nacalai Tesque; blocking solution). Sections were processed for 48 hr at 4 degrees with the primary antibody in blocking solution (1:1250, rabbit anti-Esr1, Invitrogen PA1-309), rinsed 3X10 min in PBS, then processed for 48 hr at 4 degrees with secondary antibodies in blocking solution (1:1000, Alexa Fluor 647 goat anti-rabbit, Jackson Laboratories, plus 1:500 NeuroTrace, Invitrogen). Tissue sections were then rinsed again 3X10 min in PBS, mounted on slides, and coverslipped using Fluoro-mount-G (Sothern Biotech). After drying, all the sections containing GFP labeling were imaged with a 10 X or 20 X objective on a Zeiss 700 laser scanning confocal microscope. The overlap between GFP +and Esr1 +neurons was quantified manually on all sections that contained GFP labeling (on average seven 80 µm coronal sections or 130 cells per animal), and the proportion overlap was calculated by dividing the total number of Esr1 +GFP + cells over the total number of GFP + cells from all animals.

## Transsynaptic tracing from $Amg_{C/M\text{-}PAG}$ neurons

For selective transsynaptic tracing from $Amg_{C/M\text{-}PAG}$ neurons with viruses, the caudolateral PAG was injected with AAV-pgk-retro-Cre and the $Amg_{C/M}$ was injected with a 1:1 mixture of AAV-flex-TVA-mCherry and AAV-flex-oG (total volume of 100 nL). After a wait time of 10–14 days, the $Amg_{C/M}$ was then injected with EnvA-ΔG-RV-GFP (100 nL, diluted 1:5), and animals were sacrificed after waiting an additional 4–7 days.

## Whole-cell recordings

whole-cell recordings were performed as described in *Michael et al., 2020*. Briefly, mice that received viral injections 2–4 weeks prior were deeply anesthetized with isoflurane and standard procedures were used to prepare 300-µm-thick coronal slices. The brain was dissected in ice-cold ACSF containing the following (in mM): 119 NaCl, 2.5 KCl, 1.30 $MgCl_2$, 2.5 $CaCl_2$, 26.2 $NaHCO_3$, 1.0 $NaHPO_4\text{-}H_2O$, and 11.0 dextrose and bubbled with 95% $O_2$/5% $CO_2$. The brain was mounted on an agar block and sliced in ice-cold ACSF with a vibrating-blade microtome (Leica). Slices were incubated for 15 min at 32 °C in a bath of NMDG recovery solution containing the following (in mM): 93.0 NMDG, 2.5 KCl, 1.2 $NaH_2PO_4$, 30.0 $NaHCO_3$, 20.0 HEPES, 25.0 glucose, 2.0 thiourea, 5.0 Na L-ascorbate, 2.0 Na-pyruvate, 10.0 $MgSO_4$ $7H_2O$, 0.5 $CaCl_2$, and 95.0 HCl. Slices were then moved to a bath of HEPES storage solution containing the following (in mM): 93.0 NaCl, 2.5 KCl, 1.2 $NaH_2PO_4$, 30.0 $NaHCO_3$, 20.0 HEPES, 25.0 glucose, 2.0 thiourea, 5.0 Na L-ascorbate, 2.0 Na-pyruvate, 10.0 $MgSO_4$ $7H_2O$, and 0.5 $CaCl_2$, and allowed to gradually reach room temperature over the course of 1 hr, where they remained for the duration. Recordings were performed in ACSF at a temperature of 32 °C. For voltage clamp experiments patch electrodes (4–8 MΩ) were filled with cesium internal solution containing the following (in mM): 130 cesium methanesulfonate, 5 QX-314 Br, 10 HEPES, 8 TEA-Cl, 0.2 EGTA, 4 ATP-Mg salt, 0.3 GTP-Na salt, and 10 phosphocreatine. Recordings were made using a Multiclamp 700B amplifier whose output was digitized at 10 kHz (Digidata 1440 A). Series resistance was <25 MΩ and was compensated up to 90%. Signals were analyzed using Igor Pro (Wavemetrics). Neurons were targeted using interference contrast and epifluorescence to visualize fluorescent indicators previously expressed via viral injection. ChR2-expressing axon terminals were stimulated by 5–20ms laser pulses (3–10 mW) from a 473 nm laser delivered via fiber optic inside the recording pipette (Optopatcher, A-M Systems). To confirm the direct nature of optogenetically evoked currents 2 µM TTX (Tocris) and 100 µM 4AP (Sigma-Aldrich) were added to the ACSF and perfused onto slices. To confirm that evoked currents were GABAergic, 10 µM gabazine (Tocris) was applied. Pharmacological agents including were bath applied for 10 min before making recordings.

## Code availability

Custom-written Matlab codes used in this study have been deposited to the Duke Research Data Repository, under the https://doi.org/10.7924/r4cz38d99 and are also available from the co-corresponding authors. The latest version of Autoencoded Vocal Analysis, the Python package used to generate, plot, and analyze latent features of mouse USVs, is freely available online: https://github.com/jackgoffinet/autoencoded-vocal-analysis (*Goffinet, 2021*).

## Quantification and statistical analyses

### Statistics

Parametric, two-sided statistical comparisons were used in all analyses unless otherwise noted (alpha = 0.05). No statistical methods were used to predetermine sample sizes. Error bars represent standard error of the mean unless otherwise noted. Mice were selected at random for inclusion into either experimental or control groups for optogenetic experiments. Mice were only excluded from analysis in cases in which viral injections were not targeted accurately, or in cases with absent or poor viral expression.

## Acknowledgements

Thanks to Michael Booze for mouse husbandry, thanks to Katherine Tschida for help with HCR in situ hybridization, and thanks to Fan Wang's lab for providing rabies tracing-related viruses. This work is supported by NIH grants DC 013826 (to RM) and MH 117778 (to FW and RM).

## Additional information

### Funding

| Funder | Grant reference number | Author |
|---|---|---|
| National Institute of Mental Health and Neurosciences | 5R01MH117778 | Richard Mooney |
| National Institute on Deafness and Other Communication Disorders | 5R01DC0133826 | Richard Mooney |
| National Institute on Deafness and Other Communication Disorders | 5F31DC017879 | Valerie Michael |

The funders had no role in study design, data collection and interpretation, or the decision to submit the work for publication.

### Author contributions

Shuyun Xiao, Conceptualization, Formal analysis, Investigation, Methodology, Writing – original draft, Writing – review and editing; Valerie Michael, Conceptualization, Formal analysis, Investigation, Writing – original draft, Writing – review and editing; Richard Mooney, Conceptualization, Supervision, Funding acquisition, Writing – original draft, Project administration, Writing – review and editing

### Author ORCIDs

Shuyun Xiao http://orcid.org/0009-0007-8208-7478
Richard Mooney http://orcid.org/0000-0002-3308-1367

### Ethics

This study was performed in strict accordance with the recommendations in the Guide for the Care and Use of Laboratory Animals of the National Institutes of Health. All experiments were conducted according to protocols approved by the Duke University Institutional Animal Care and Use Committee protocol (# A227-17-08).

### Decision letter and Author response

Decision letter https://doi.org/10.7554/eLife.85547.sa1
Author response https://doi.org/10.7554/eLife.85547.sa2

## Additional files

### Supplementary files

• MDAR checklist

## Data availability

Data are deposited to the Duke Research Data Repository. There are 3 types of data in the repository: (1) confocal microscope images, (2) audio and video files from the mice used in this study and (3) slice electrophysiology data. All other data analyzed in this study are included in the manuscript and supporting files.

The following dataset was generated:

| Author(s) | Year | Dataset title | Dataset URL | Database and Identifier |
|---|---|---|---|---|
| Xiao S, Michael V, Mooney R | 2023 | Data from: Nested circuits mediate the decision to vocalize | http://doi.org/10.7924/r4x35002b | Duke Research Data Repository, 10.7924/r4x35002b |

The following previously published dataset was used:

| Author(s) | Year | Dataset title | Dataset URL | Database and Identifier |
|---|---|---|---|---|
| Michael V, Goffinet J, Pearson J, Wang F, Tschida K, Mooney R | 2020 | Data and scripts from: Circuit and synaptic organization of forebrain-to-midbrain pathways that promote and suppress vocalization | https://doi.org/10.7924/r4cz38d99 | Duke Research Data Repository, 10.7924/r4cz38d99 |

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
