## [Editor Report]

Vocalizations are controlled by neural circuits connecting the amygdala and periaqueductal gray. This study presents valuable measures of the neurons that suppress vocalization in appropriate contexts using a rich variety of behavioral, imaging, optogenetic, and tracing methodologies. The evidence supporting the claims of the authors is solid, and their results provide synaptic mechanisms by which vocal-promoting and vocal-suppressing signals could interact in the mouse's brain to underlie the hierarchical control of vocalization. The work will be of interest to neurobiologists working on motor control and vocalization.

---

## [Decision Letter]

**Decision letter after peer review:**

Thank you for submitting your article "Nested circuits mediate the decision to vocalize" for consideration by *eLife*. Your article has been reviewed by 2 peer reviewers, and the evaluation has been overseen by a Reviewing Editor and Andrew King as the Senior Editor. The reviewers have opted to remain anonymous.

Essential revisions:

1. For the optogenetic stimulation of POA Amg neurons, the data presented only included 5 male mice. Please provide information about the data from the females used in these experiments.

2. Provide more details on the POA Amg cells, and compare with data from previous studies that elicit vocalizations by stimulating POA PAG.

3. Your idea that the reproductive and social benefits of vocalizing may outweigh the potential costs of advertising one's location to predators and aggressive social rivals is very interesting. Could you address this in more detail?

4. Please provide more detail about how your experiments were performed to address the comments made in particular by reviewer #2.

*Reviewer #1 (Recommendations for the authors):*

1) In lines 555-558 in the Methods, "For optogenetic stimulation of POA Amg cells the Amgc/m of C57 male and female mice were injected with AAV-pgk-retro-Cre (100nL) and the medial POA was injected with AAV-flex-ChR2 (100nL) and the optogenetic ferrule was placed above the POA Amg cell bodies. However, the data presented only included 5 male mice. What about the females in these experiments?

2) If possible, I would like more details on the POA Amg cells since they are a subpopulation of POA PAG cells. What proportion of these make up the population of POA PAG cells examined in Michael et al., 2020? When compared to the previous study eliciting vocalizations by stimulating POA PAG, how do the results compare? The success of eliciting vocalizations, latency, and acoustics?

3) To test if 'the asymmetry of this circuit supports a logic where the reproductive and social benefits of vocalizing may outweigh the potential costs of advertising one's location to predators and aggressive social rivals' (Lines 510-514), the pieces of a potential experiment exist in the manuscript. In addition to eliciting vocalization in the absence of social cues via stimulation of POA Amg cells, would stimulation in the presence of 2MT or while the mouse is actively being attacked also elicit vocalizations? Can you override the suppression of AMGc/m-PAG neurons to show this putative asymmetry?

---

## [Author Response]

Essential revisions:1. For the optogenetic stimulation of POA Amg neurons, the data presented only included 5 male mice. Please provide information about the data from the females used in these experiments.

We thank reviewers for catching this typo in the Methods. For the reasons outlined in the preamble above, this section focuses only on male mice. We have revised the methods (line 803) to say “For optogenetic stimulation of POA Amg cells the Amgc/m of C57 male mice were injected with AAV-pgk-retro-Cre (100nL) and the medial POA was injected with AAV-flex-ChR2 (100nL) and the optogenetic ferrule was placed above the POA Amg cell bodies.”

We specified this male-centric focus in the section title (line 521) “In male mice, POA neurons that provide input to the Amg_C/M_ are Esr1+ and optogenetically activating them is sufficient to promote USV production.”

We added texts explaining our rationales for why we focused on only male mice in the second part of the study (experiments described in Figures 4-6) in the introduction (lines 93-95), results (lines 419-435) and Discussion sections (lines 663-670). We also added text discussing the limitations to the current study in the Discussion section, stating that whether optogenetic stimulation of POA-Amg neurons is sufficient to elicit vocalizations remains to be tested in females (lines 722-729).

2. Provide more details on the POA Amg cells, and compare with data from previous studies that elicit vocalizations by stimulating POA PAG.

We have added a panel of three figures to Figure 6 (Figure 6H) comparing results from current study to those from Michael et al., 2020 (Figure 2F). USVs elicited by optogenetically stimulating POA_Amg_ neurons (current study; dark blue) did not differ significantly from those elicited by optogenetically stimulating (1) POA_PAG_ neurons, (2) Esr1+ POA neurons or (3) Esr1+ POA terminals at PAG in terms of mean USV rate (Kruskal-Wallis test, Chi-sq = 6.36, df = 3, *p* = 0.0954), success rate (Kruskal-Wallis test, Chi-sq = 5.1, df = 3, *p* = 0.1645), and mean latency (Kruskal-Wallis test, Chi-sq = 7.12, df = 3, *p* = 0.0681) (statistics included in legend for Figure 6H, lines 589-603).

3. Your idea that the reproductive and social benefits of vocalizing may outweigh the potential costs of advertising one's location to predators and aggressive social rivals is very interesting. Could you address this in more detail?

We have expanded the treatment of this idea in the revised discussion (lines 671-691; 713-722). We have also added Figure 7 (line 692) to illustrate the model of nested circuits that mediate the decision to vocalize described by the current study.

4. Please provide more detail about how your experiments were performed to address the comments made in particular by reviewer #2.

Our responses have been provided below for each comment by reviewer #2. Experimental details have been added to figure legends and methods accordingly.

Reviewer #1 (Recommendations for the authors):1) In lines 555-558 in the Methods, "For optogenetic stimulation of POA Amg cells the Amgc/m of C57 male and female mice were injected with AAV-pgk-retro-Cre (100nL) and the medial POA was injected with AAV-flex-ChR2 (100nL) and the optogenetic ferrule was placed above the POA Amg cell bodies. However, the data presented only included 5 male mice. What about the females in these experiments?

We thank reviewers for catching this typo in the Methods. For the reasons outlined in the preamble above, this section focuses only on male mice. We have revised the methods (line 803) to say “For optogenetic stimulation of POA Amg cells the Amgc/m of C57 male mice were injected with AAV-pgk-retro-Cre (100nL) and the medial POA was injected with AAV-flex-ChR2 (100nL) and the optogenetic ferrule was placed above the POA Amg cell bodies.”

We specified this male-centric focus in the section title (line 521) “In male mice, POA neurons that provide input to the Amg_C/M_ are Esr1+ and optogenetically activating them is sufficient to promote USV production.”

We added texts explaining our rationales for why we focused on only male mice in the second part of the study (experiments described in Figures 4-6) in the introduction (lines 93-95), results (lines 419-435) and Discussion sections (lines 663-670). We also added text discussing the limitations to the current study in the Discussion section, stating that whether optogenetic stimulation of POA-Amg neurons is sufficient to elicit vocalizations remains to be tested in females (lines 722-729).

2) If possible, I would like more details on the POA Amg cells since they are a subpopulation of POA PAG cells. What proportion of these make up the population of POA PAG cells examined in Michael et al., 2020? When compared to the previous study eliciting vocalizations by stimulating POA PAG, how do the results compare? The success of eliciting vocalizations, latency, and acoustics?

We have added a panel of three figures to Figure 6 (Figure 6H) comparing results from the current study (dark blue) to those from Michael et al., 2020 (light blue; source data from Figure 2F in Michael et al., 2020). USVs elicited by optogenetically stimulating POA_Amg_ neurons (current study; dark blue) did not differ significantly from those elicited by optogenetically stimulating (1) POA_PAG_ neurons, (2) Esr1+ POA neurons or (3) Esr1+ POA terminals at PAG in terms of mean USV rate (Kruskal-Wallis test, Chi-sq = 6.36, df = 3, p = 0.0954), success rate (Kruskal-Wallis test, Chi-sq = 5.1, df = 3, p = 0.1645), and mean latency (Kruskal-Wallis test, Chi-sq = 7.12, df = 3, p = 0.0681). Error bars show S.D.

3) To test if 'the asymmetry of this circuit supports a logic where the reproductive and social benefits of vocalizing may outweigh the potential costs of advertising one's location to predators and aggressive social rivals' (Lines 510-514), the pieces of a potential experiment exist in the manuscript. In addition to eliciting vocalization in the absence of social cues via stimulation of POA Amg cells, would stimulation in the presence of 2MT or while the mouse is actively being attacked also elicit vocalizations? Can you override the suppression of AMGc/m-PAG neurons to show this putative asymmetry?

These are interesting experiments but limits on time and resources prevent us from conducting them. However, we previously showed that the POA > PAG pathway promotes vocalizations in male and female mice (Michael et al., 2020). Here we showed that stimulation of POA>PAG axon collaterals in the Amg can also promote vocalizations in male mice. Lastly, we show here that male and female mice will vocalize to female partners in the presence of 2MT, even though they do so less frequently. We maintain that this provides a more naturalistic test of the idea that POA/PAG/Amg cells can override 2MT, which we show here activates the vocal suppressing cells in the Amg and suppresses USV production. We acknowledge that the explicit test of whether optogenetic stimulation of POA_PAG_>Amg collaterals can promote USV production in the presence of vocal-suppressing stimuli is lacking and agree that future experiments are needed to test the full dynamic range of interactions between vocal-promoting and vocal-suppressing afferents to the amygdala and the PAG (we added this point to our discussion of limitations in lines 724-729). However, we also maintain that our present efforts provide novel insights into this process and are well suited to the Research Advance format, as opposed to a new Article.